# Aging, inflammation and DNA damage in the somatic testicular niche with idiopathic germ cell aplasia

Massimo Alfano [1,7✉], Anna Sofia Tascini [2,7], Filippo Pederzoli [1,3], Irene Locatelli[1], Manuela Nebuloni [4], Francesca Giannese[2], Jose Manuel Garcia-Manteiga [2], Giovanni Tonon [2], Giada Amodio [5], Silvia Gregori[5], Alessandra Agresti [6✉], Francesco Montorsi[1,3] & Andrea Salonia [1,3]

Molecular mechanisms associated with human germ cell aplasia in infertile men remain undefined. Here we perform single-cell transcriptome profiling to highlight differentially expressed genes and pathways in each somatic cell type in testes of men with idiopathic germ cell aplasia. We identify immaturity of Leydig cells, chronic tissue inflammation, fibrosis, and senescence phenotype of the somatic cells, as well markers of chronic inflammation in the blood. We find that deregulated expression of parentally imprinted genes in myoid and immature Leydig cells, with relevant changes in the ratio of Lamin A/C transcripts and an active DNA damage response in Leydig and peritubular myoid cells are also indicative of senescence of the testicular niche. This study offers molecular insights into the pathogenesis of idiopathic germ cell aplasia.

[1] Division of Experimental Oncology/Unit of Urology, URI, IRCCS Ospedale San Raffaele, Milan, Italy. [2] Center for Omics Sciences, IRCCS Ospedale San Raffaele, Milan, Italy. [3] Università Vita-Salute San Raffaele, Milan, Italy. [4] Pathology Unit, Department of Biomedical and Clinical Sciences, L. Sacco Hospital, Università degli Studi di Milano, Milan, Italy. [5] San Raffaele Telethon Institute for Gene Therapy (SR-TIGET), IRCCS Ospedale San Raffaele, Milan, Italy. [6] Division of Genetics and Cell Biology, IRCCS Ospedale San Raffaele, Milan, Italy. [7] These authors contributed equally: Massimo Alfano, Anna Sofia Tascini. ✉email: alfano.massimo@hsr.it; agresti.alessandra@hsr.it

Spermatogenesis depends on the presence of testicular spermatogonial stem cells, along with their amplification and transition into progenitor, differentiating spermatogonia and, eventually, spermatozoa[1]. Germ cell aplasia (GCA), also known as Sertoli cell-only syndrome (SCOS) from a histological point of view, represents the most severe form of male infertility, clinically defined as non-obstructive azoospermia (NOA).

Although both monogenic and oligogenic mutations are associated with NOA[2], in almost 80% of NOA cases it is not possible to identify a clear cause, thus including those cases in the wide group of idiopathic NOA (iNOA)[3]. In this context, several studies demonstrated the relevance of the somatic cells on stem cell number and male spermatogenesis in the testicular microenvironment[4–7]. Similarly, the testis microenvironment has a central role in GCA[8], including an altered vitamin A and vitamin K signaling in Sertoli and Leydig cells, respectively[9], and an enriched testicular bacterial load[10].

Epidemiological data show that infertility status per se, and mostly in NOA cases, is associated with an increased risk of early onset of age-related chronic diseases (e.g., cardiovascular disease, type II diabetes mellitus, autoimmune disease, and cancers)[11,12] with a relatively higher risk of death[11] compared to fertile men. Overall, these data support the concept that male infertility may be considered as a proxy of overall men's health[13].

Here, we leveraged the single-cell RNAseq to unveil the occurrence of dysregulated transcriptional pathways in testicular somatic cell populations in idiopathic GCA (iGCA). Findings from this study depict the contribution of 7 somatic cell populations to the complex framework of iGCA, and provide a list of defects potentially associated with an early onset of age-related pathological processes in men with iNOA. Moreover, most of the transcriptional signatures have been confirmed at tissue and systemic levels. Overall, the translational outcomes of this study have a relevant impact on the recognition of a number of biological processes and biomarkers associated i) with male infertility, and ii) early onset of aging processes in young men still in their reproductive age.

## Results

**Single-cell transcriptome profiling revealed seven somatic cell clusters in iGCA.** In iNOA men with negative sperm retrieval at microdissection TEsticular Sperm Extraction (microTESE), the presence of seminiferous tubules lacking any germ cell configured the histological diagnosis of GCA (Fig. 1a). Conversely, the histological analysis of a case of obstructive azoospermia (OA), associated with cystic fibrosis transmembrane conductance regulator (CFTR) gene mutation, showed a condition of normal spermatogenesis, with sperm cells at different stages of maturation in the seminiferous tubules (Fig. 1a).

To investigate the molecular pathogenesis of iGCA and provide a comprehensive description of the testicular somatic cell populations, we performed single-cell RNA-sequencing (scRNA-seq) on freshly isolated tissue specimens from microTESE. A total of 3880 cells, obtained after filtering out poor-quality cells from 3 iGCA donors (Supplementary information, Fig. S1a–1d, Dataset S1), were integrated to reduce batch effects, and the Seurat standard graph-based clustering approach was used for cell partitioning and cluster identification. Overall, 8 cell clusters were identified as shown in the Uniform Manifold Approximation and Projection (UMAP)[14] plot (Fig. 1b). Using previously determined cell type marker genes[15,16] (Supplementary information, Dataset S2), the main somatic cell populations were identified, Leydig (LEY), myoid (MYD), Sertoli (SRT), and endothelial (END) cells (Fig. 1c; Supplementary information, Fig. S1e–1j, Dataset S1). Furthermore, immune cells were also identified, such as macrophages (MCR) and T-cells (TCL) (Fig. 1c; Supplementary information, Fig. S1e–1j). Based on the expression of the marker genes RGS5 and TPM2 we assigned the stromal (STRO) cluster to pericytes[17,18] or vascular smooth muscle cells according to the expression of FABP4[19,20](Fig. 1c; Supplementary information, Fig. S1j), which represent progenitors stem cell of LEY cells[21]. The relative abundance of somatic cell populations (Supplementary information, Fig. S1k), along with the lack of germline markers expression (Supplementary information, Fig. S1l), confirm the absence of germ cells at any stage throughout spermatogenesis in men with iNOA. Of all, one cluster of cells remained undetermined (UND) (Fig. 1b), since it lacked clear maker genes and had a low number of RNA features (Fig. 1d; Supplementary information, Fig. S1m).

To depict the unique characteristics of each somatic cell population in iGCA testis, we applied functional enrichment analysis to each cluster (Fig. 1d; Supplementary information, Dataset S3–10). Pathways specific for LEY and MYD cells included "collagen biosynthesis" and "extracellular matrix (ECM) organization". SRT cells were enriched in the "pregnenolone biosynthesis" pathway, the first substrate along the steroid hormones synthesis pathway[22]. In MCR and TCL groups we found highly enriched pathways related to activation of the immune system that include "cytokine-mediated signaling", and "PD-1 signaling" pathways. Finally, in END and STRO cells, "vasculogenesis" and "extracellular matrix organization" pathways were enriched, respectively.

**Single-cell transcriptomics of somatic cell clusters in normal spermatogenesis.** In parallel, scRNA-seq of testicular tissue of one man with OA associated with CFTR gene mutation was used as a positive control for cell cluster identification (Supplementary information, Fig. S2a, Dataset S1). By applying the same cell partitioning protocol used for iGCA samples, we also analyzed published data from three autoptic testicular tissues with normal spermatogenesis found in a published study[15] which reported data for 6490 cells, 1676 of which were somatic (Supplementary information, Fig. S2b, Dataset S11). After cluster identification, OA cell types were assigned based on the expression of somatic and germ cell marker genes, according to the consolidated descriptions of testicular cell type[15]. Cells identity was also characterized by projecting the PCA structure of the reference dataset from Guo and colleagues[15] onto both OA and iGCA datasets. The unsupervised graph-based clustering approach and the projection of reference data led to a similar outcome in terms of cluster identity, with the majority of cells isolated from OA testis being ascribed to the spermatogenic pathway (Fig. 1e; Supplementary information, Fig. S2a), while only somatic cells were found in iGCA testis (Fig. 1f).

**Cell-specific upregulated pathways in somatic cell populations associated with germ cell aplasia.** To unravel differences in terms of somatic testicular environment, we compared gene expression in each somatic population from iGCA testes and from testes with normal spermatogenesis, respectively. Data from OA testis have not been used as a reference for normal tissues since the number of somatic cells in this sample was low and not all the niche populations were fully represented; moreover, in OA patients bearing CFTR mutations, there is overexpression of genes in the inflammatory response pathway[23] that might affect the enrichment analyses. To enable comparison and exclude batch effects, the datasets from our 3 iGCA testes were integrated with those from the five testicular samples with normal spermatogenesis retrieved from two different datasets, Guo[15] and Sohni[24]. Cell partitioning was performed as described above, and cluster

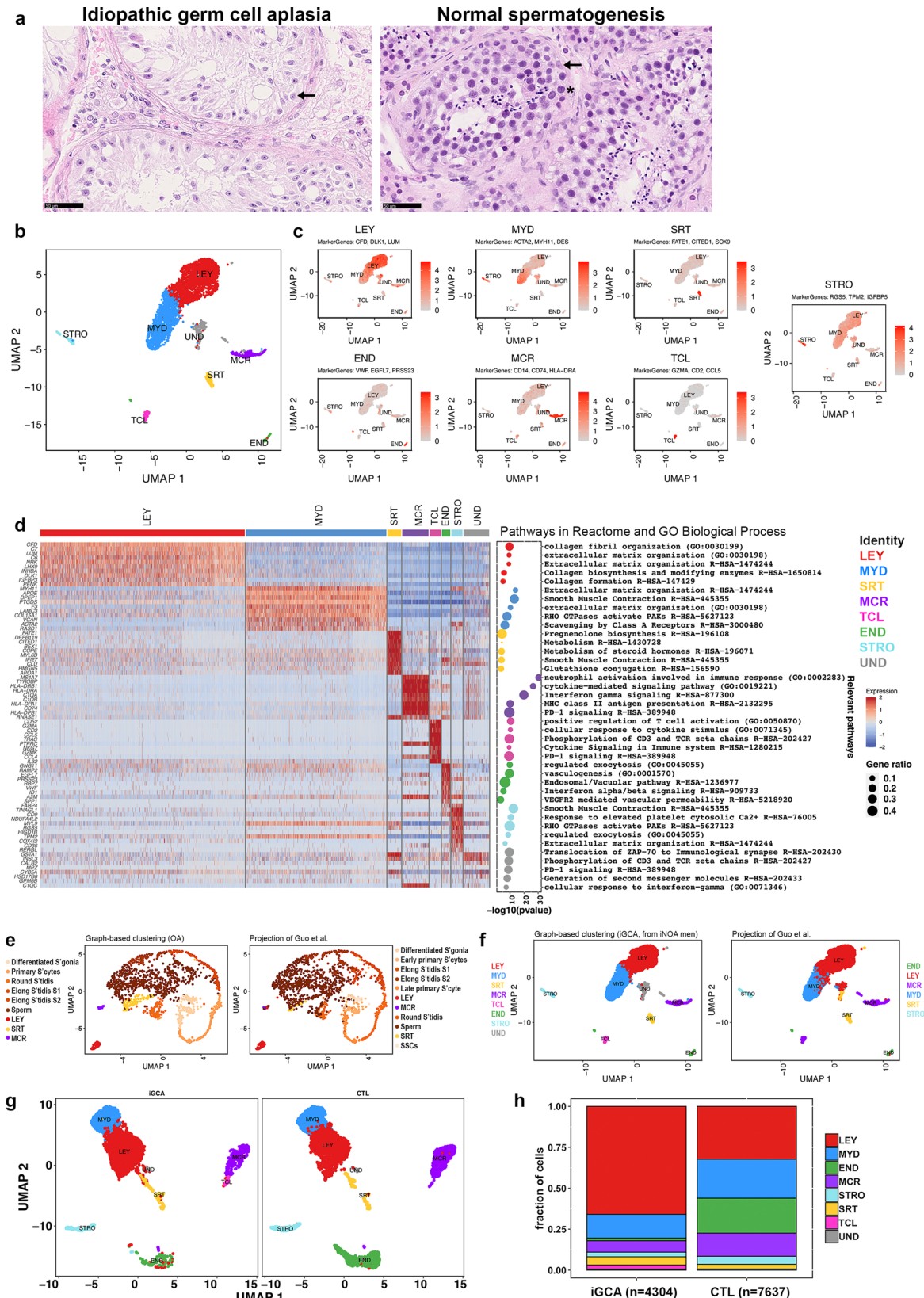

identity was assigned based on the expression of somatic cell marker genes (Supplementary information, Fig. S3, Dataset S12), with 7 somatic cell types clearly identified (Fig. 1g). We observed that the relative abundance of the somatic cell populations varied between iGCA and samples with normal spermatogenesis, and the TCL cluster was only present in the iGCA testes (Fig. 1h).

Functional enrichment analysis was used to identify those gene ontologies and pathways upregulated in testicular somatic cells from iGCA (Supplementary information, Fig. S4, Dataset S13–19). As for MYD, LEY, and STRO cells, the top upregulated pathways were related to ECM deposition and organization. Further upregulated pathways common to MYD and LEY cells were also

**Fig. 1 Clustering of somatic cells from iGCA through the analysis of marker genes expression. a** Histological analysis of testis parenchyma with complete iGCA (left panel; one donor representative of five tested with the similar result) and normal spermatogenesis (right panel; one donor representative of five tested with the similar result): the arrow indicates SRT cells characterized by a dense nucleolus; the asterisk indicates the spermatocytes. **b** UMAP plot representing testis cells from tissue samples of three independent iNOA men with complete iGCA (replicates in Supplementary information, Fig. S1). **c** UMAP plots showing the expression patterns of selected marker genes used to identify the different cell types (Supplementary information, Dataset S1, 2). **d** Left: Heatmap showing the expression signature of the top ten expressed genes in each cell type. Right: representative pathways for each cell type derived from the Reactome Pathway Database and GO Biological Process (Supplementary information, Dataset S3–10). **e** Comparison of the unsupervised clustering of cells from the testis of OA man (Supplementary information, Dataset S1) and cell identity prediction by Data Transfer on a reference (see M&M), provided by the three samples from Guo et al.[15]. **f** Comparison of the unsupervised clustering of cells from the testis with iGCA (Supplementary information, Dataset S1) and cell identity prediction by Data Transfer on a reference, provided by the three samples from Guo et al.[15]. **g** UMAP plots resulting from the integration of cells from the testis with iGCA and selected somatic cells from 5 testes with normal spermatogenesis from Guo et al.[15] and Sohni et al.[24] (Supplementary information, Fig. S3, Dataset S12). **h** Relative abundance of all somatic cell populations based on gene markers from three testes with iGCA and five control testes with normal spermatogenesis (CTL). Source data are provided as a Source Data file.

"Cholesterol biosynthesis" and "Scavenging by Class A Receptors (SR-A)", which included the overexpression of *SCARA5* in LEY cells as previously reported[25]. "Antigen presentation" and "Pre-NOTCH transcription and translation" pathways were upregulated in MYD cells, while "Nuclear-transcribed mRNAs catabolic process" and "Cellular response to corticosteroid stimulus" were specifically upregulated only in LEY cells (Fig. 2a–c). Upregulated pathways in SRT cells were related to energy metabolism, including the glyceraldehyde-3-phosphate metabolic process (Fig. 2d).

The analysis of resident immune cells revealed a drift toward a pro-inflammatory phenotype. In MCR we found upregulated pathways related to MHC class II antigen presentation, interferon-gamma, and NF-kB signaling (Fig. 2e; Supplementary information, Fig. S4). CD8+/CD69+ T-cells were identified in iGCA testes but not in the datasets of control testes from Guo[15] and Sohni[24] (Fig. 2f). In addition, projecting the iGCA transcriptome onto a dataset of T cells isolated from human tissues[26] confirmed the phenotype of the T-cells from iGCA testis as tissue-resident CD8 + cytotoxic T cells (Fig. 2g). Furthermore, CD8 + CD69 + TCL expressed granzyme K and M, the pro-inflammatory chemokines CCL4/MIP1β and CCL5/RANTES (Fig. 2h) and cytokine IL-32 (Supplementary information, Dataset S12), supporting the conclusion that these somatic tissue-resident cytotoxic T cells might be generated by a previous inflammatory or autoimmune insult and contribute to chronic local inflammation. All TCL in the iGCA were cytotoxic T cells; conversely, a recent dataset on human testes with normal spermatogenesis showed the presence of a few T-cells, of which only 23% of expressed CD8[27].

To investigate the differences in cellular communication between iGCA and healthy donor, we performed an unbiased ligand–receptor interaction analysis between these testicular cell subsets by CellphoneDB[28]. Overall, we found that the total number of interactions was significantly increased in iGCA donors, especially among MCR, LEY, and MYD cells (Fig. 2i, Supplementary information, Dataset S20). The increased interaction was prompt by the overexpression of the cell surface receptor CD74 in iGCA MCR (Fig. 2j, k).

**Cell-specific downregulated pathways in somatic cell populations associated with iGCA.** Fifty-six downmodulated pathways were common to all somatic cell populations in iGCA, with the aforementioned exception of the TCL cluster (Fig. 3a; Supplementary information, Fig. S5, Dataset S21–28). Most of those pathways were related to protein metabolism and ribosomal biogenesis. In all somatic iGCA populations, we also found downregulation of the selenocysteine synthesis and selenoamino acid metabolism pathways. In line with these findings, we identified the downregulation of *INMT*, an enzyme involved in

selenium metabolism and detoxification[29] in LEY cells, and the glutathione peroxidases *Gpx1* and *Gpx4* (Fig. 3b) in SRT cells. Since selenium is also involved in redox detoxification of the extracellular milieu, we wondered whether also redoxin gene family was differentially expressed. The transcripts for several selenoproteins were downregulated in all iGCA somatic cells (Fig. 3b; Dataset S29), including the *SELENOP* (i.e., the most important and abundant selenoprotein in plasma with extracellular antioxidant functions) and *SELENOH* (which has redox capabilities linked to genome damage and DNA stress and a nucleolar localization)[30].

Other downregulated pathways of particular interest were the mitochondrial respiratory chain complex and the oxygen-dependent proline hydroxylation of hypoxia-inducible factor (Fig. 3c; Supplementary information, Dataset S29).

Yet not exclusively, the common pathways were mainly guided by significant downregulation of ribosomal protein genes expression across all cell types. Indeed, a differential expression of ribosomal genes has recently been reported in the somatic cells of fetal and adult human testis and associated with a differentiated metabolic activity[24]. To exclude the presence of potential artifacts due to different library preparations, we quantified the ribosomal genes expression global levels in samples from our cohort and from the Guo[15] and Sohni[24] datasets; overall, ribosomal protein genes were overexpressed in somatic cells compared to germ cells and the read coverage onto ribosomal protein genes was comparable in our OA sample and the ones from Guo and Sohni, thus pointing to a real biological difference rather than batch effects (Supplementary information, Fig. S6a).

The same up and downregulated pathways identified by comparing somatic cells from iGCA vs. CTL testis were identified through the comparison of iGCA vs. the testis with obstructive azoospermia (Supplementary Fig. S6b–e).

**Transcriptional and phenotypic immaturity state of LEY cells in iGCA.** Paracrine cell-cell signaling is involved in the control of germ cell development and spermatogenesis[31]. Based on our previous work, showing that iNOA men present with higher levels of anti-mullerian hormone (AMH)[32] and altered biochemical ECM composition associated with impaired metabolism of the retinoic acid[9], we hypothesized that the somatic testicular niche of iGCA might be representative of a testis stuck at an immature stage. Here, we provided several observations to support our hypothesis. First, we examined the expression of the Delta Like Non-Canonical Notch Ligand 1 (*DLK1*) gene, which is considered a marker of the immaturity of LEY cells. *DLK1* is expressed in the human testis during fetal, neonatal, and prepubertal stages[24,33], is an inhibitory modulator of Notch signaling[34], whose block determines the final maturation of fetal

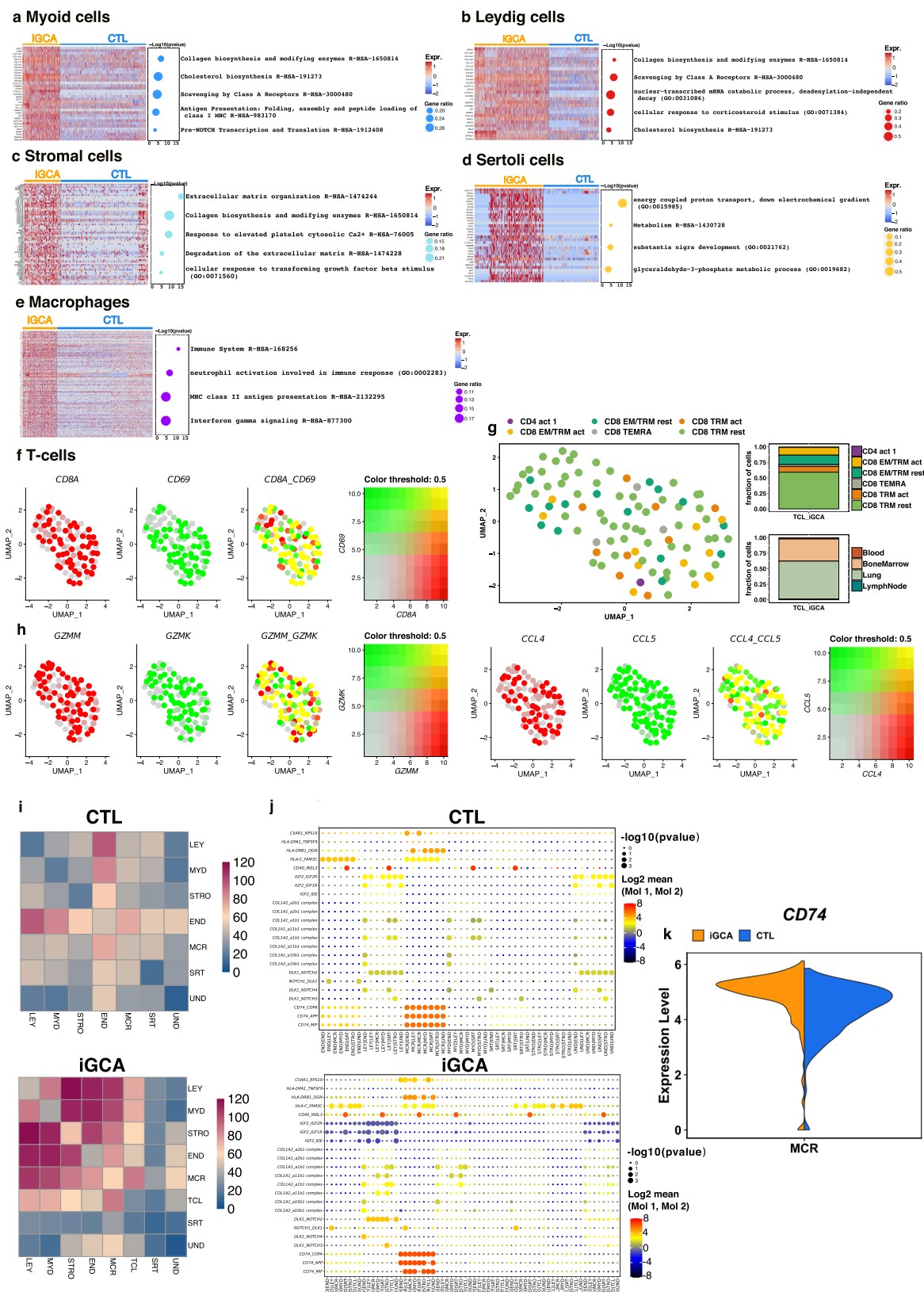

LEY cells[35]. Here, we found overexpression of *DLK1* and *NOTCH2* in LEY and MYD cells in iGCA vs. adult testis (Fig. 4a, Supplementary information, Dataset S29 and validation at the protein level in Supplementary Fig. S7). Along with the immature transcriptional phenotype of LEY cells, we considered the IGF/ IGFR pathway whose signaling on adult SRT and LEY cells[36] is directly responsible for the testicular size and sperm production[37,38] and for steroidogenic activity[39], respectively. We found that *IGF1* was expressed in LEY and MYD cells, and *IGF2* down-modulated in LEY, MYD, and SRT cells in iGCA (Fig. 4b; Supplementary information Fig. S8), while no modulation of the relative receptors *IGF1R* and *IGFR2* was observed.

**Fig. 2 Upregulated pathways in myoid, leydig, stroma cells, and macrophages associated with iGCA, together with the presence of T cells in the testis with iGCA and cell–cell interactions.** Heatmap of differentially expressed genes that contributed to the upregulated pathways in **a** MYD cells, **b** LEY cells, **c** STRO cells, **d** SRT cells, and **e** MCR of iNOA men with GCA vs. control tests with normal spermatogenesis (CTL), as described by the Reactome Pathway Database and GO Biological Process (Supplementary information, Fig. S4, Dataset S13–15); the color scale of the heatmap represents expression values, and the heatmap shows the intensity of gene expression per each cell. Dot plots summarize the most relevant pathways, and the dot size is proportional to the percentage of genes expressed in each pathway. **f** The blend plot visualizes the co-expression of the *CD8* and *CD69* antigen transcripts in the T cell (TCL) population (left). **g** The projection map and the relative distribution of iGCA TCL identities defined on the classification of Szabo et al.[26]. **h** Blend plots visualizing the expression of functional markers as *GZMM*, *GZMK* (left), *CCL4*, and *CCL5* (right) in the TCL of iNOA men with GCA (Supplementary Dataset S12). **i** Heatmap showing the total number of interactions between cell types in the decidua dataset obtained with CellPhoneDB for IGCA samples (right) and healthy donor (right). **j** DotPlot of selected ligand–receptor interactions produced using CellPhoneDB. P values are represented by increasing circle size. The means of the average expression level of interacting molecule 1 in cluster 1 and interacting molecule 2 in cluster 2 are indicated by color. **k** Violin plot for CD74 in iGCA MCR vs. control adult testis. Source data are provided as a Source Data file.

Overall, *DLK1* overexpression and decreased *IGF2* expression underpin two main pathways that might advocate for LEY cell immaturity. INSL3 is a marker for mature LEY cell[40] and we here report lower expression of INSL3 at both gene and protein levels in iGCA LEY cells vs. adult testis (Fig. 4c). Transcripts for *HSD17B3*, an enzyme that converts androstenedione to testosterone and is associated with virilization[41], were present with lower levels of expression only in a fraction of LEY cells (Fig. 4d; Supplementary information, Dataset S29), further supporting the conclusion of a delayed LEY cells maturation in iGCA. Cairns group[15,42] recently reported differentiation trajectories for LEY cells based on the analyses of single-cell RNA seq from neonatal, prepubertal, and adult human testes (Supplementary information, Fig. S9, Dataset S30). We extracted from Cairn's datasets the transcriptional signatures regarding the early, intermediate, and late stage of LEY cell differentiation (A, B, and C, respectively) and compared them with the transcriptional profiles of iGCA LEY cells. We report that iGCA LEY has a greater expression of the Stage A and B signatures, typical of the pre-pubertal maturation stage, while the controls have a greater expression of the Stage C signature, which is typical of the post-pubertal maturation stage (Fig. 4e). The signature for LEY immaturity included the overexpression of *EGR1* (Supplementary information, Fig. S9f), which has been reported to mediate inflammation and fibrosis[43,44], and of *JUNB*, recently proposed as a marker gene of SRT cell immaturity[19]. Furthermore, available single-cell datasets from multiple studies[15,24,42,45], showed that *DLK1* expression in LEY and MYD follows a decreasing trend from the neonatal to the prepubertal and adult stages. Consistent with these findings, *DLK1* expression in iGCA was similar to what was observed in the neonatal testis (Fig. 4f). Finally, *INSL3* expression was found to increase from the neonatal to adult age, but in iGCA LEY the expression was similar to that of LEY cells from the neonatal testis (Fig. 4f).

**Deregulated expression of paternally and maternally imprinted genes.** Starting from the observation that both *DLK1* and *IGF2* are paternally imprinted genes, we investigated the genome-wide transcriptional landscape of all the other imprinted genes (http://www.geneimprint.org/site/what-is-imprinting). By comparing the integrated datasets from the Guo[15] and Sohni[24] with ours, we found that imprinted genes are dramatically either over or underexpressed in LEY and MYD populations associated with iGCA, (Fig. 4g; Supplementary information, Fig. S10a, b, Dataset S31, 32). Even more important, several imprinted genes are highly expressed in the vast majority of LEY and MYD cells from iGCA testis while completely absent in control cells. The same holds true for genes absent in iGCA cells and expressed in CTL cells. Unexpectedly, a common signature of biallelic expression in iGCA somatic cells was found for some imprinted genes, such as IGF2 (Fig. 4h), with some genes imprinted in all three iGCA and

some others specific for each iNOA men (Supplementary Dataset S33) indicating genome-wide deregulation of imprinting molecular mechanisms.

**Inflammatory signature at the tissue level.** As for the upregulated pathways related to ECM, collagen formation, and collagen organization, we found that COL1A1 and COL1A2 were overexpressed in LEY and MYD cells in iGCA (Fig. 5a). The levels of collagen I and collagen IV proteins, the main types of collagen forming the basal membranes, were quantified by immunehistochemical analyses, using testicular specimens from men with iNOA men depicting iGCA and an age-matched internal cohort of men with normal spermatogenesis. As a further control, we also included testicular specimens from iNOA men with severe hypospermatogenesis, where germ cell-devoid seminiferous tubules intermingled with tubules positive for sperm cells at microTESE.

Thin layers of collagen I were detected around the seminiferous tubules in men with normal germline maturation while tubules from iNOA patients with either complete GCA or hypospermatogenesis showed thick accumulation of collagen I (Fig. 5b). Conversely, collagen type IV was present at the basal membrane of seminiferous tubules with normal germline maturation and absent in iNOA patients with either complete GCA or hypospermatogenesis (Fig. 5c). Of interest, collagen type IV deposition was normal at the basal membrane of the vessels, regardless of the presence of seminiferous tubules with or without sperm production (Fig. 5c).

The median value of basal membrane thickness of 4, 6, and 20 μm was observed in the testis with normal spermatogenesis, iNOA with hypospermatogenesis, or iGCA, respectively (Fig. 5d). We found that the increased thickness of the basal membrane in the tubules was associated with a steady decrease in sperm retrieval (Fig. 5e). In terms of collagen organization, we observed alignment of collagen fibrils in the basal membrane of tubules with normal spermatogenesis, but a scattered distribution in the tubules with iGCA (Fig. 5f), thus resulting in a lower degree of collagen fibrils organization (Fig. 5g). These findings highlighted a pathologic modulation of collagen I and IV both at transcriptional, protein, and organizational levels in MYD and LEY as compared with END cells environment. The "leopard pattern" hyalinization of the seminiferous tubules is indicative of a potential spreading of the disease.

The deregulated collagen deposition at the basal lamina as a feature of tissue aging added to inflammatory features and the detection of both T-cell and activated macrophages prompted us to evaluate a possibly active innate inflammatory response in iGCA testis. We analyzed the cellular localization of HMGB1, the prototypical nuclear danger-associated molecular pattern (DAMP) protein[46], which at first relocates to the cytoplasm, and then is secreted in the extracellular milieu as a danger signal

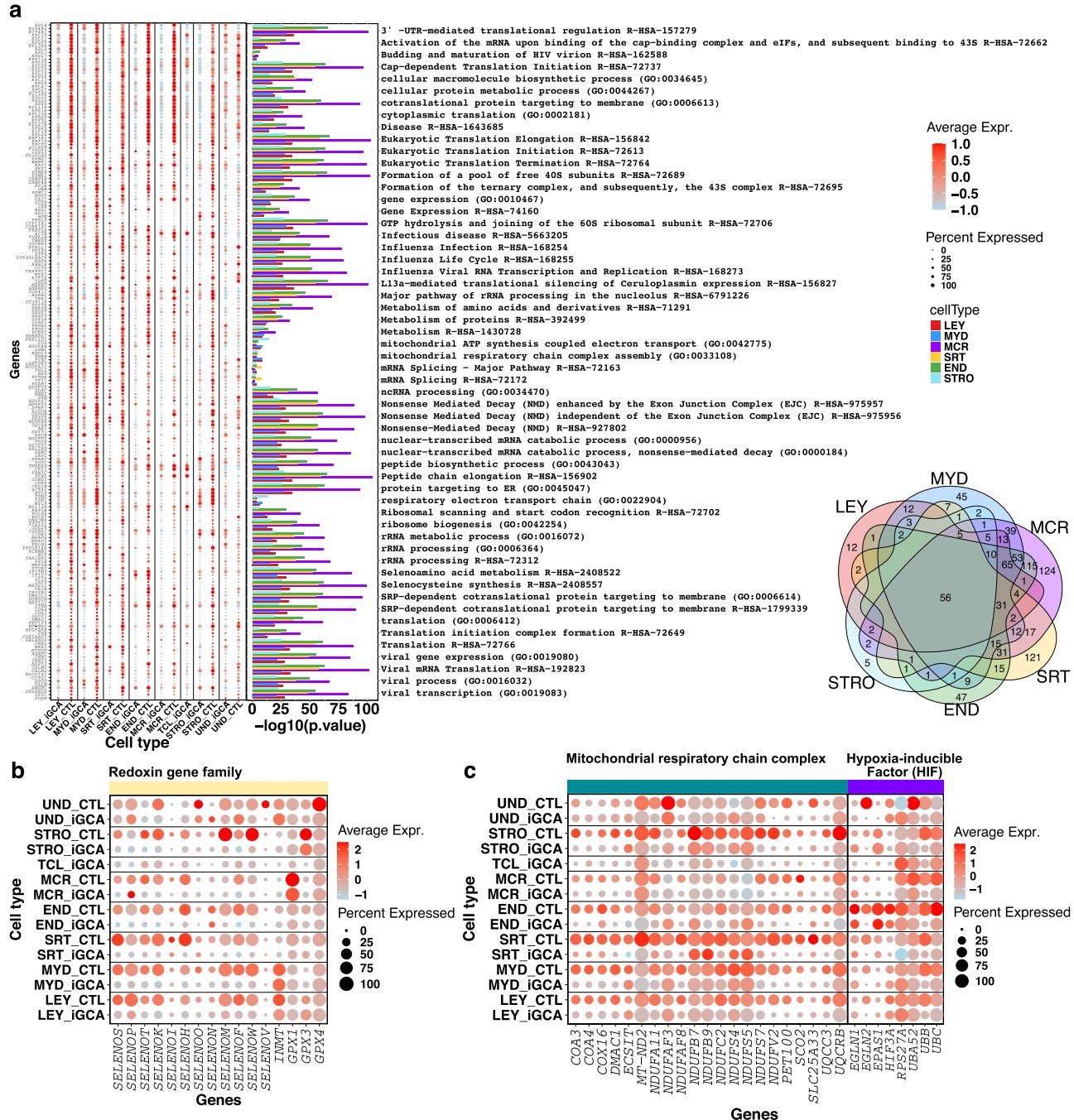

**Fig. 3 Downregulated pathways shared by all the somatic cells of men with iGCA. a** Dot plot comparison of shared downregulated genes and pathways in the seven somatic cell populations in iGCA. Only genes downregulated in at least two cell types are considered. Pathways are derived from the Reactome Pathway Database and GO Biological Process and only those common in all cell types are displayed (Supplementary information, Fig. S5, Dataset S21-27). The Venn diagram shows the intersection of the 56 down-modulated pathways in the six somatic cell populations of iNOA men with germ cell aplasia (Supplementary information, Fig. S5, Dataset S28). **b** Dot plot comparison of gene expression for enzymes involved in Selenium metabolism and Redoxin proteins (Supplementary information, Dataset S29). **c** Dot plot comparison of gene expression for proteins involved in the pathway "Mitochondrial respiratory chain complex" and "Hypoxia-inducible factor". Source data are provided as a Source Data file.

to neighboring cells upon inflammatory insults. HMGB1 also belongs to the large group of senescent activated secretory phenotype proteins (SASP)[47], whose secretion correlates with senescence. Somatic cells in three non-neoplastic testes (median age 34, interquartile ranges (IQR) 29–36), showed nuclear HMGB1 localization with a similar degree of staining. Conversely, HMGB1 was found in the cytoplasm of SRT, LEY, and MYD cells in all seven iGCA tested (Fig. 6a). These results

indicate that the somatic fraction of iGCA testis cells is actively responding to an inflammatory environment.

**DNA damage and structural alteration of the nuclear envelope in somatic cells.** The higher tumor frequency reported for men with azoospermia prompted us to proceed from chronic tissue inflammation to senescence and DNA damage as possible causes of genomic instability in iGCA.

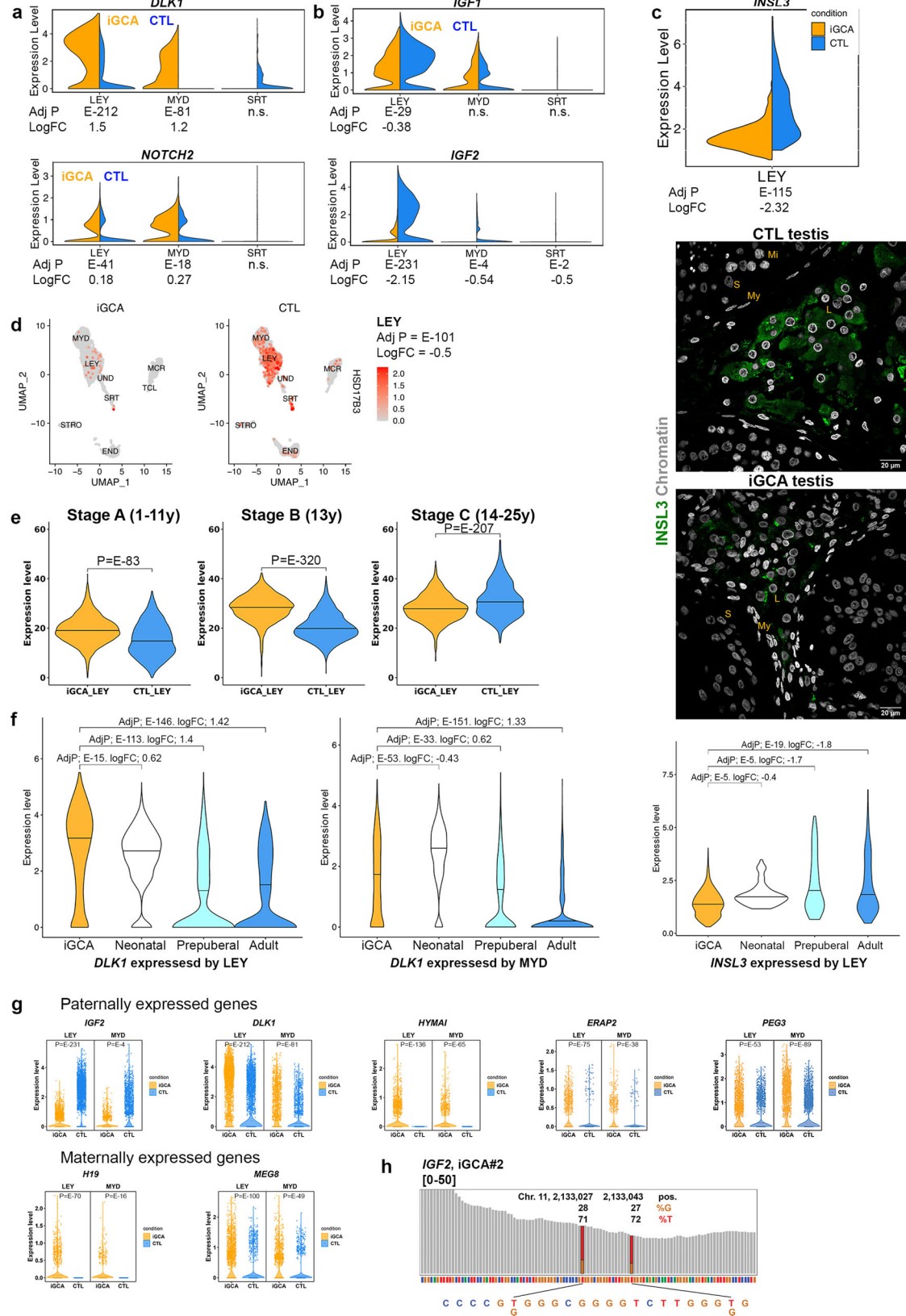

To support the senescent phenotype inferred by SASP detection, we looked for the presence of known indicators of cellular senescence-like cell cycle-related proteins. Indeed, although p16/CDKN2A (Cyclin-Dependent Kinase Inhibitor 2A) is not differentially expressed in iGCA somatic cells, the staining revealed a strong accumulation of p16 in iGCA LEY cells as opposed to control testis (Supplementary Fig. S11a).

SASP is also linked to chronic DNA damage response (DDR) that is detected by the nuclear presence of phosphorylated histone H2AX (γH2AX), which is considered an early marker of DNA

**Fig. 4 Immature phenotype of Leydig cells in germ cell aplasia.** Violin plots for *DLK1* and *NOTCH2* (**a**), and *IGF1* and *IGF2* (**b**) expression in iGCA vs. the somatic cells of control adult testis. **c** Violin plot for *INSL3* transcription in iGCA vs. control adult testis, and INSL3 protein staining in testis parenchyma (one CTL testis and one iGCA testis representative of four tested each condition, with similar result). Leydig cells are recognized by their localization amongst tubules and the dense chromatin ring at the nuclear periphery, while Sertoli cells are recognized for the presence of a characteristic DAPI negative condensed nucleolus. Mi meiotic cells, S Sertoli cells, My peritubular myoid cells, L Leydig cells. **d** Feature plots of *HSD17B3* transcripts in iGCA and control testis (Guo and Sohni datasets); DGE in Supplementary information, Dataset S30. **e** Violin plots compare expression levels of Stage A–C signatures of LEY cell maturation in iGCA vs. healthy donor (two-sided Mann Whitney test). Signatures were established by combining three datasets reporting single-cell RNA seq of neonatal, prepubertal, and adult human testis analysis detailed in Supplementary Fig. S9). **f** *DLK1* and *INSL3* expression in iGCA vs. neonatal, prepubertal, and adult control testis. **g** Violin plots compare the expression of imprinted genes in the somatic cell clusters in iGCA and controls (two-sided Mann Whitney test). **h** One representative imprinted gene (*IGF2*) with biallelic expression in the three tested iGCA testis. Reads distribution from the iGCA#2 scRNA-seq are reported for a portion of the *IGF2* gene locus (NM_000612.6, exon 4). Gray bars indicate a perfect match of the sequence reads at the nucleotide level with the human GRCh38/hg38 reference sequence, bar height indicates the coverage. The "red and orange" bars indicate the positions where 27–28% of reads diverge from the reference sequence (pos. 2,133,027 and 12,133,043), thus suggesting a biallelic expression. Adj P adjusted p value, two-sided Mann Whitney test followed by Bonferroni correction for multiple comparisons. LogFC log2 fold change. Source data are provided as a Source Data file.

damage[48]. By immunofluorescence, DNA damage was observed in the cell line treated with $H_2O_2$ (Supplementary Fig. S12) and meiotic chromosomes in spermatocytes (leptotene/pachytene) of control testis showed the expected dotted nuclear staining for γH2AX[49] (Fig. 6b). SRT, MYD, and LEY nuclei were negative in the control testis but stained positive in iGCA tissues (Fig. 6b, up to 35% of positive cells in each population, $n = 3$), thus pointing to an active DDR. In addition, by DAPI staining, we noted very distorted nuclei in the LEY and SRT cell population, reminiscent of nuclei in patients with nuclear envelopathies, a group of rare genetic disorders characterized by a variety of clinical symptoms connected with premature aging and caused by mutations in genes encoding for components of the nuclear lamina[50]. Indeed, the membrane of LEY nuclei from iGCA, but not SRT nuclei, were significantly more indented than the CTL cell counterparts ($P < 0.001$) as assessed by the nuclear circularity parameter, whereby the value 1 corresponds to a smooth nuclear surface and 0 to high levels of nuclear indentation (Fig. 6c). Likewise, both LEY and SRT nuclei in iGCA showed a significantly lower degree of roundness, a parameter that indicates a shift toward an oval shape as opposed to the roundish shape detected in CTLs (Fig. 6c).

Consistent with the pathogenesis of envelopathies, we found a mutation in the *lamin A/C* gene (LMNA, ex.10, H566H, C > T, c.1698; rs4641, the allelic frequency of approx. 0.2, depending on the population considered[51]) in our three iGCA patients which were in homozygosity (TT) for iGCA#2 and in heterozygosity (C/T) for iGCA #1 and #3; the same mutation was not observed in the 5 CTL datasets (Fig. 6d), by manually inspecting the scRNA-seq reads accumulated on the *LMNA* locus. So far, this base transition has been considered a synonym polymorphism due to the absence of pathologic phenotypes besides very mild associations with metabolic syndromes[52,53]. No mutations in other envelope genes were detected, either.

Of note, the rs4641 LMNA polymorphism hits the −1 position of the splicing donor (SD) site at the 3′ end of exon 10 (Supplementary Fig. S13) which rules the delicate balance between the *A* and *C* alternatively spliced forms.

Since the wildtype *LMNA* gene bears the infrequent SD sequence "AC" in ex10, which has a higher affinity for the U1 spliceosome complex compared to the consensus "AG" SD (half-life = 1.34 vs. 1 a.u., respectively[54]), we hypothesized that the T substitution in the mutant alleles would (i) lead to a shorter U1 spliceosome complex half-life (1.12 a.u.), (ii) decrease the splicing events probability at that SD site, and (iii) both decrease the production of the full-length *LMNA* transcript and favor the truncated form *LMNC*. Indeed, the *kallisto*[55] isoform analysis applied to the high-coverage RNA-seq on the *LMNA/C*

locus (3000–6000 reads) identified five major *LMNA/C* transcript isoforms (Fig. 6d) in CTLs and iGCA cases. *LMNA* and *LMNC* transcripts are reduced by almost twofold in iGCA as compared to CTLs (Var1 and Var2; $p = 0.018$ and 0.01, respectively) while a significant fivefold enrichment of Variant 5 was detected ($p = 0.02$). The overexpression of Variant 5 functionally supports the hypothesis of the weak splice donor site: indeed, the transcript terminates with the exon10 sequence plus a short sequence from intron 10 (18 bp plus stop codon and 3′UTR in intron 10), which codes for the same 6aa present in *Variant 2/LMN C*. These data suggest that exon10-to-exon11 splicing events have lower odds in iGCA with rs4641 *LMNA* polymorphism than in CTLs with the wildtype gene splice sequence. In addition, the 5′end is transcribed from an alternative ex1, which excludes the first 100 aa in the protein that is then replaced by an alternative unrelated 20 aa sequence (Fig. S13).

Overall, our results in iNOA patients phenotypically support a *p16/CDKN2A*+ senescent phenotype in LEY cells that are cell cycle blocked and with high levels of DNA damage to deal with. In addition, the genetic mutation in the LMNA gene in iGCA cells might drive an unbalanced representation of Lamin protein isoforms at the nuclear envelope determining misshapen nuclei reminiscent of those from aged individuals and progeric children[56].

**Features of inflammation in iGCA at a systemic level.** A further step was to investigate if the above reported transcriptional findings are reflected at the systemic level. Indeed, the upregulation of pregnenolone biosynthesis transcriptional pathway identified in iGCA SRT cells (Fig. 1d) was paralleled by an increased peripheral level of estradiol in 41 iNOA men with iGCA (median age 37 years; IQR 33–39) vs. 41 normozoospermic men (median age 37 years; IQR 34–41) with proven fertility (Fig. 6e).

We also found decreased levels of circulating total testosterone (tT) in the 41 iNOA men with GCA vs. the 41 normozoospermic individuals with proven fertility (Fig. 6f). These data further support the concept that LEY immaturity contributes to a general and systemic phenotype. Circulating tT and luteinizing hormone (LH) undergo an age-dependent gradual shift in males, with a decrease of tT along with an LH increase[57]. In a cohort of 102 fertile men (median (IQR) 37 (33–40) years) we observed that LH/tT ratio was independent of sperm parameters, whereas the same ratio was increased in an age-matched cohort of iNOA men with GCA ($n = 43$; 37 (33–39) years) (Fig. 6g). Therefore, low levels of tT potentially associated with an immature status of LEY cells were associated with compensatory positive feedback in terms of LH synthesis.

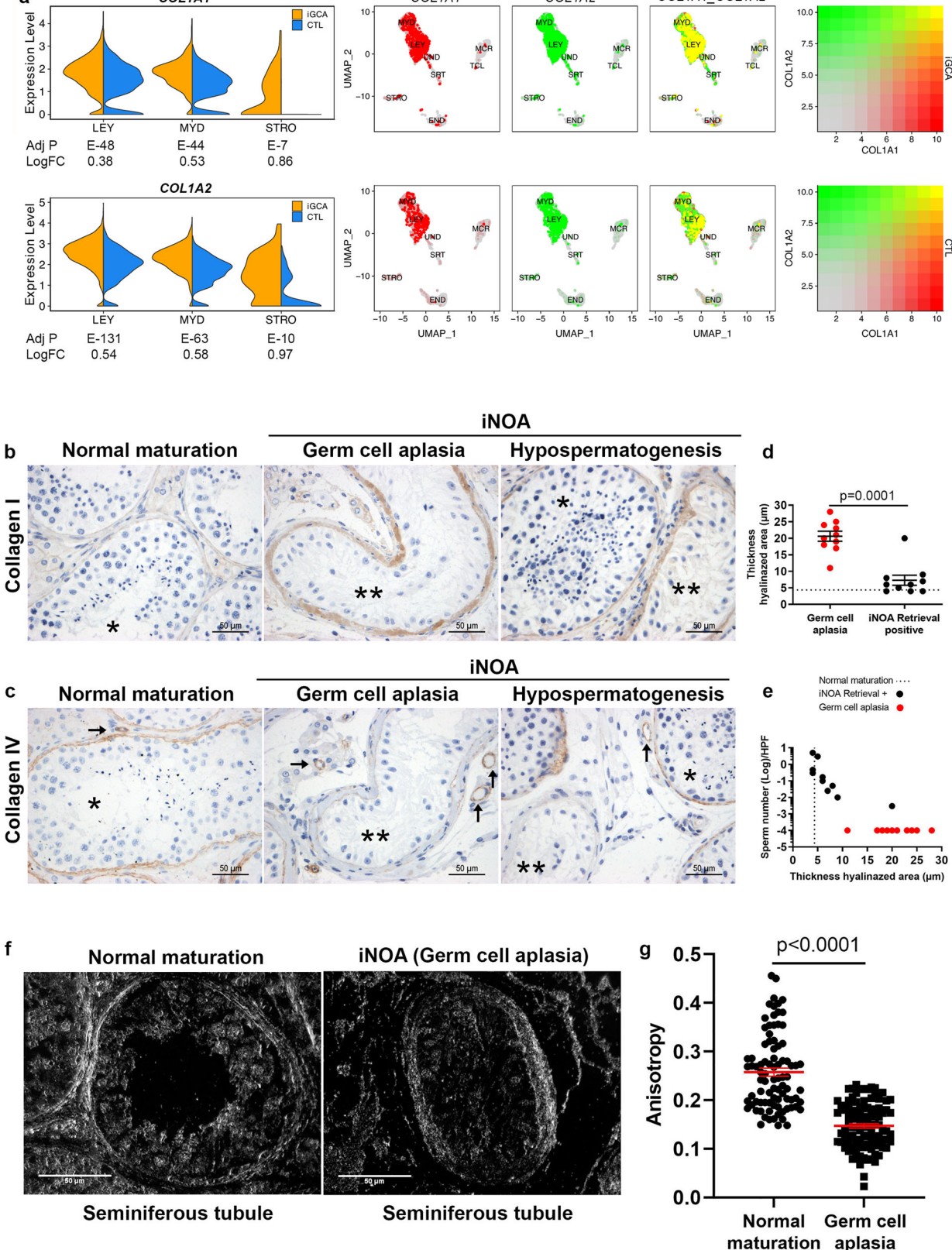

The deficiency of testosterone, known to inhibit the production of inflammatory cytokines[58], associated with the cytoplasmic localization of HMGB1, the pro-inflammatory profile of MCR, and the presence of tissue-resident CD8+ TCL in iGCA patients, prompted us to investigate whether the inflammatory processes are only localized in the testis environment or are more systemic.

Prompted by the overexpression of cytokines and chemokines in TCL, we quantified blood circulating CCL4 and CCL5, which belong to the SASP proteins secreted in inflammation and senescence. In iNOA men with iGCA vs. age-matched controls, levels of CCL4 and CCL5 chemokines are significantly higher (Fig. 6h, i).

**Fig. 5 Overexpression of collagen I and down-modulation of collagen IV in the basement membrane of seminiferous tubules with iGCA. a** Violin plots compare the expression levels of collagen genes in LEY, MYD, and STRO cells. Blend graphs show the degree of expression and co-expression in the somatic populations from iGCA (upper panels) and CTLs (lower panels). AdjP adjusted P value; LogFC logarithmic fold change iNOA vs. CTL. n.s. not significant. **b** Deposition of type I collagen and **c** type IV collagen in the basal membrane of seminiferous tubules and vessels in the case of normal spermatogenesis, iGCAor hypospermatogenesis (one testis for each condition representative of five tested for each condition, with similar result); * = seminiferous tubules with spermatogenesis; ** = seminiferous tubules iGCA; arrow = vessels. **d** Thickness values of the thickness of the hyalinized area surrounding the seminiferous tubule were quantified (ImageJ software version 1.5 applied on high magnification images). A single dot represents the average value of twenty seminiferous tubules from each clinical specimen. Ten independent testes with germ cell aplasia (red dots) and ten from iNOA and positive sperm retrieval patients (black dots) were analyzed; the dotted line shows the average value for five independent normozoospermic samples, each with 20 seminiferous tubules counted; error bars represent mean ± SEM. Statistical significance was evaluated by means of two-tail Mann–Whitney. **e** Sperm retrieval was expressed as the number of sperm/high power field (HPF) from testis specimen with hypospermatogenesis; the association between sperm retrieval and the thickness of the collagen area is reported. **f** Images produced by collagen birefringence in the basal membrane of the seminiferous tubules with normal spermatogenesis and iGCA (one testis for each condition representative of five tested for each condition, with the similar result). **g** Quantification of the degree of collagen fibrils organization (anisotropy) in the basal membrane of 100 tubules with normal spermatogenesis ($n = 20$ tubules each donor, $n = 5$ independent donors) and with iGCA ($n = 20$ tubules each donor, $n = 5$ independent donors); red bars represent mean ± SEM; two-sided Mann Whitney test. Scale bar; 50 μM. Adj P Bonferroni adjusted p value for multiple comparisons, LogFC log2 fold change, iNOA idiopathic non-obstructive azoospermia. Source data are provided as a Source Data file.

**iGCA testis show a transcriptional aged phenotype**. As a whole, the upregulated proteins of the innate immunity and cytokine signals, a pro-inflammatory environment, p16 staining, low levels of circulating testosterone, along with the downregulated pathways of RNA processing and amino acids metabolism, reported for iNOA men, are consistent with altered biological pathways orchestrating the process of ageing[59]. Comparative analysis of Differentially Expressed Genes vs. the aging-associated genes annotated in the GenAge database (https://genomics.senescence.info/legal.html) clearly revealed an overlap with all ribosomal genes, *UBA52* and *NACA* reported being associated with aging, for all seven somatic iGCA clusters (Fig. 6j).

## Discussion

Our work provides a comprehensive transcriptional atlas of the testicular somatic cell populations associated with human iGCA. We further identified deregulated features at transcriptional, tissue and systemic levels in iGCA patients thus providing a detailed picture on the immaturity of LEY cells, senescence of the testicular somatic cells, defects in gene imprinting, and ongoing inflammation with an activated senescent secretory phenotype. Overall, this study lays the ground for a better understanding of the pathogenesis of iGCA.

Overall, historically testicular somatic cells have long been regarded for their support to life-long spermatogenesis. Of those, SRT cells are known to provide structural support and nutrients to germ cells, and peritubular MYD cells to secrete factors important for spermatogenesis, while providing contractile function to the seminiferous tubules. Likewise, LEY cells produce testosterone and secrete factors that regulate the functions of both SRT and MYD cells, along with the immunological activities carried out by MCR[37].

Previous findings indirectly suggested that SRT cells in testes with iGCA are stuck at the pre-puberal stage[9,32]; recently, a maturation arrest of SRT cells in men with iNOA has been confirmed, along with increased energy metabolism and estradiol production[19]. Here, we confirmed these latter findings on Sertoli cells of iGCA testes. We further identified LEY, MYD, and STRO cells as the main somatic contributors to altered collagen type I and IV deposition and ECM disorganization associated with iGCA. The unbalanced distribution of collagen type I and IV in the basal membrane of the seminiferous tubule was strictly associated with the lack of germ cells. Thanks to the axial tissue sections, we highlighted a peculiar scattered collagen distribution with a "leopard pattern" in the basement membrane of adjacent tubules whose interpretation is twofold. It could be representative of a homogeneous peculiar feature in all the tubules or be the evidence of a heterogeneous condition due to the presence at the same time of few tubules with a partially preserved spermatogenesis and tubules with GCA. Both conditions are potentially indicative of progressive disease, and the translational implication might be iNOA to become a preventable disease if fertility protocols were implemented. Moreover, the transcriptional profiles of END cells in iGCA identified a collection of modulated pathways related to aging, also shown by all the other six somatic cell populations, a phenotype that is poorly compatible with their instructive and supportive role toward other testicular somatic cell types and germ cells[37].

We further reported a significant nuclear deformation and indentation in both LEY and SRT cells in iGCA testes that were not detected in CTLs and clearly reminiscent of envelopathies in aged individuals and children with genetic mutations in genes encoding for nuclear Lamin proteins[56]. Genetic envelopathies determine cardiomyopathies, metabolic and progeric syndromes with degenerative disorders, which collectively overlap frequent pathologies in iGCA men.

Lamin A/C (LMNA/C) is one of the principal components of the envelope and one of the most studied genes since it harbors hundreds of relevant mutations, many of which are linked to the Hutchinson–Gilford syndrome, an extremely rare, progressive genetic disorder that causes children to age rapidly.

From the molecular analyses in the three iGCA transcriptomes, we identified an excess of shorter LMNC-like transcripts that we attributed to a mild polymorphism in the *LMNA/C* gene and results in a disproportionate meshwork of intermediate filament type V in the nuclear envelope. The lamin isoforms ratio is cell and tissue-specific and is designed to architecture chromatin in lamin-associated domains, which eventually governs cell and tissue-specific transcription[60]. In addition, the nuclear envelope also sustains nuclear mechanics and mechanotransduction[61]. Therefore, structural misshapen nuclei might be directly responsible for dysfunctional transcriptional landscapes in iGCA or, vice versa, harsh environments might induce structural damage on otherwise weak nuclei followed by dramatic transcriptional changes. In addition, prolonged mechanic and inflammatory stresses might sustain the overload of DNA damage, which eventually leads to genomic instability and cancer[62]. In addition, an aberrant nuclear organization can also lead to gene imprinting defects as recently demonstrated[63]. Our findings are just adding some more pieces to the puzzle of germ cell aplasia and will be worth deeper explorations in the future.

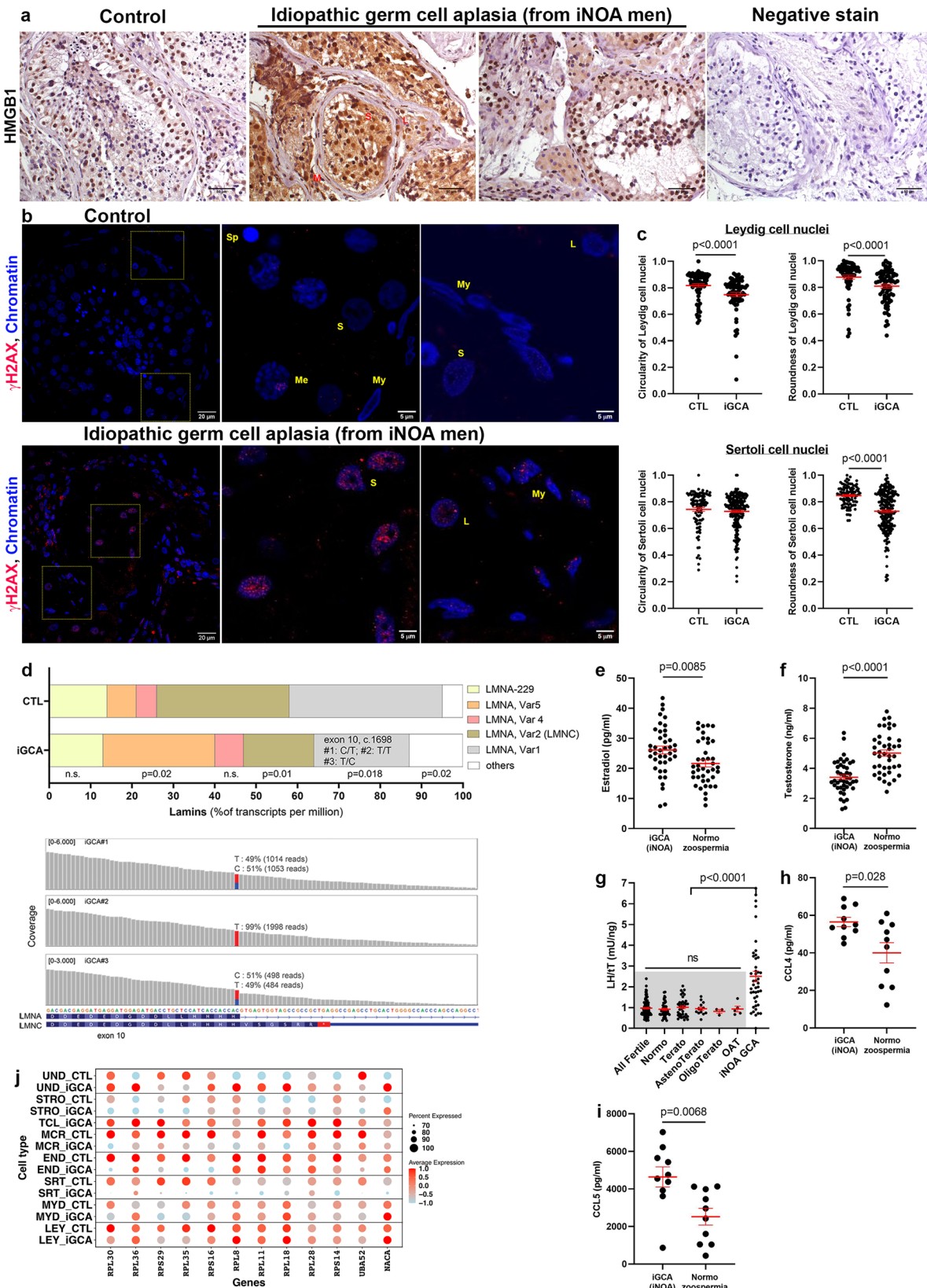

During the development, the somatic cells reconstitute the genomic imprinting present in the zygote. Of note, several paternally and maternally imprinted genes are massively upregulated or downregulated in LEY and SRT cells in our iGCA patients as compared to CTLs (Fig. 6). The observation that those genes are located on several different chromosomes (i.e., 5, 6, 11,

14, and 19), together with the unexpected biallelic expression of some pairs, expands the limited involvement of gene-specific mechanisms[64–66] to a genomewide defect recalling multi-locus imprinting disturbances[67] as opposed to faulty methylation of specific regions, worth to be characterized in the future. Any metabolic or environmental factors during embryo development

**Fig. 6 Inflammation and aging markers at the local and systemic level. a** Localization of HMGB1 (brown stain) in 1 out of 3 representative testes with normal spermatogenesis and in 2 out of 7 representative testes with iGCA; blue letters indicate cluster of Leydig cells (L), Sertoli (S), and peritubular Myoid (My) cells. **b** Distribution of γH2AX (red signal) in one representative sample of three testes with normal spermatogenesis and one representative of three testes with iGCA; low magnification of the seminiferous tubules are followed by higher magnification to appreciate γH2AX dots localization in Meiotic cells (Me), but not sperm cells (Sp) or somatic cells in the control testis, and in SRT, MYD, and LEY cells from iGCA patient. **c** Comparisons of circularity and roundness values as proxies for nuclear envelope deformation in LEY and SRT cells; LEY cells were identified by immunofluorescence staining of lineage-specific marker CALB2 (Supplementary Figs. S7 and S11) and the latter by intratubular localization ($n = 4$ independent donors both for CTL and iGCA, for a total of 98 SRT and 67 LEY cells in CTL and 214 SRT and 76 LEY cells in iGCA). Red bars represent mean ± SEM; two-sided Mann Whitney test. **d** Distributions of the most represented *Lamin A/C* RNA isoforms detected in the sc-RNAseq datasets. Specific transcript fractions are reported as percentages over total LMNA/C transcripts (tpm) in normal (Guo plus Sohni datasets, total controls = 5) and iGCA samples (iGCA#1–3). The polymorphism in the *LMNA* exon ten donor splice site is reported for each iGCA subject. **e** Peripheral levels of estradiol and **f** testosterone in men with iGCA and normozoospermic men with proven fertility (n = 41 independent donors). Red bars represent mean ± SEM; two-sided Mann Whitney test. **g** LH/Tt ratio in fertile men (n = 102 independent donors) clustered according to sperm parameters into normozoospermia = 43; teratozoospermia = 38; asteno-teratozoospermia = 13, oligo-teratozoospermia = 3, oligo-asteno-teratozoospermia = 5 and in iNOA men with germ cell aplasia (GCA; n = 43 independent donors). Red bars represent mean ± SEM. tT total testosterone, OAT oligo-asteno-teratozoospermia. Two-sided Krustal–Wallis test corrected for the false discovery rate (procedure of Benjamini, Krieger, and Yekutieli). **h**, **i** Comparison of the distribution of CCL4 and CCL5 chemokines blood concentration (n = 10 independent donors for both iGCA and normozoospermia); red bars represent mean ± SEM. **j** Comparison of expression levels for ribosomal protein genes among the somatic clusters in testis with iGCA and healthy controls. iNOA idiopathic non-obstructive azoospermia. Source data are provided as a Source Data file.

might also account for imprinting alterations in iGCA somatic cells. Indeed, gestational hypertension/eclampsia is found associated with changes in the methylation levels of *DLK1*[68] and *IGF2*[69,70], which we also find substantially deregulated; moreover, offspring born from preeclamptic women are at higher risk for metabolic diseases in later life (e.g., hypertension, stroke, and obesity), conditions that apply also to iNOA men[71,72]. These findings provide hints on the epigenetic etiopathogenesis behind iGCA and offer relevant working hypotheses for an early onset of age-related chronic diseases in azoospermic men.

The increased signaling of IFNs in TCL and MCR cells, documents a shift from the immune-privileged status[73] to a pro-inflammatory testicular environment, eventually associated with GCA. In addition, the identified resident cytotoxic TCL, type 1 MCR, and the hyalinization of the seminiferous tubules support the hypothesis of a previous inflammatory or autoimmune event which turned into a chronic inflammatory status. However, the detection of cytoplasmic and extracellular HMGB1, a prototypical DAMP protein secreted upon inflammatory insults and inflammatory stimulus per se[74], supports an active inflammatory status of the testes in iGCA patients.

It is well-known that a chronic proinflammatory environment predisposes to increased incidence of cancer[75]. Several findings in this study corroborate the setting to a pre-neoplastic environment in iGCA tissues, such as the biallelic and deregulated expression of the imprinted gene IGF2, reported in more than 50% of human testicular germ cell tumors[76] and the accumulation of γH2AX in MYD and LEY cells indicative of a sustained DDR.

Some deregulated metabolic pathways in iGCA testes might also contribute to preneoplastic environments and oncogenes activation[77]. For instance, the oxygen-dependent proline hydroxylation downregulated pathway—a process that has been linked to stabilization and nuclear translocation of HIF1-alpha—and the downregulated transcription of selenium metabolism genes impair the redox control in tissues and blood (SELENOP). In addition, SELENOH proteins and the Gpx family of peroxir-eductases contribute to the redox status control and to the maintenance of genome stability[78].

Uncontrolled redox stress in immature LEY cells, in the overall iGCA somatic compartment, and in blood hint to processes related to global early aging.

This study unveils insights into idiopathic germ cell aplasia, such as the immature phenotype of LEY cells stuck at the pre-puberal stage of development and expressing clear markers of

senescence that might support the hypothesis of exhaustion in the somatic progenitor cell compartment, still needing an experimental validation.

Pathways linked to mechanisms potentially responsible for the condition of iGCA open perspectives for a better understanding of the pathobiology behind the clinical condition of iNOA and the potential association with a number of clinical manifestations, such as chronic non-oncological diseases and higher rates of cancers.

## Methods

**MicroTESE, sperm retrieval, and tissue processing.** Specimens and complete clinical data were collected at one tertiary-referral center for reproductive medicine. According to the World Health Organization (WHO) criteria, infertility was defined as not conceiving a pregnancy after at least 12 months of unprotected intercourses[79]. Men were classified as azoospermic when no spermatozoa were detected in at least two consecutive semen analyses[80]. Idiopathic non-obstructive azoospermia was defined after a comprehensive investigation of all known causes of azoospermia, according to the inclusion and exclusion criteria recently reported[32].

Adult human testicular samples: scRNA-seq was performed with fresh testicular tissue obtained from four azoospermic men during microTESE. Of those, three men were characterized by iNOA with negative sperm retrieval at microTESE and a histological diagnosis of SCOS; #1 (37 years; FSH = 36 mUI/ml; LH = 10.84 mUI/ml; tT = 2.69 ng/ml), #2 (32 years; FSH = 27 mUI/ml; LH = 7 mUI/ml; tT = 2.06 ng/ml), #3 (41 years; FSH = 26.8 mUI/ml; LH = 12.4 mUI/ml; tT = 1.93 ng/ml). Testicular tissue specimen with normal spermatogenesis was obtained from 1 man with obstructive azoospermia (age = 37 years) associated with CFTR gene mutation, which was positive for sperm retrieval at surgery and depicted normal spermatogenesis at the histological analysis (37 years; FSH = 7.4 mUI/ml; LH = 4.09 mUI/ml; tT = 6.14 ng/ml).

All four azoospermic men underwent microTESE according to the original surgical technique[81]. Sperm retrieval was attempted on testis specimen, and patients were then dichotomized according to the surgical outcome (namely, positive vs. negative sperm retrieval at microTESE); tissue was delivered to the Unit of Obstetrics/Gynecology and processed for sperm retrieval as a part of in vitro fertilization program; sperm were counted, and sperm retrieval was expressed as the number of sperm/high power field (HPF)[32]. ScRNA-seq data from adult human testis were obtained from two previous studies[15,24], which used testes removed from three deceased men (age = 17, 24, and 25 years), and from two vasectomies (age = 37 and 42) respectively, and proven to have normal spermatogenesis at the histological analysis[15].

Testicular specimens dedicated to this study were delivered from the operating theater to the laboratory on cold Krebs solution (Merck KGaA); after being split, part of the tissue was fixed for histological analysis, while the remaining part was processed for single cell dissociation. Normal maturation of the germline was defined according to a thorough histological analysis performed on formalin-fixed paraffin-embedded slides of the testis.

The expression of collagen I, collagen IV, and HMGB1 were analyzed through immunohistochemistry (IHC) staining performed on FFPE slides of the testis according to a methodology previously described[82]. Briefly, three μm paraffin

sections from each sample were pretreated in a microwave oven, two cycles of 5 min each, at 780 W in 0.25 mM EDTA buffer, and then incubated for 2 h with specific antibodies against human antigens and counterstained with hematoxylin. The reactions were revealed by a non-biotin peroxidase detection system with 3,3-diaminobenzidine free base as the chromogen. Primary antibodies against human antigens were anti-Collagen I (1:500 dilution, Ab34710 from Abcam, Cambridge, UK), anti-Collagen IV (1:100 dilution, M0785 from Dako, Denmark), anti-HMGB1 (Abcam 18256); negative control was obtained by omitting the primary Ab. Control tissues with normal spermatogenesis used for IHC analysis were obtained from the non-neoplastic tissue of men submitted to unilateral orchiectomy for nonmetastatic seminoma; the tissue was obtained from the most distant area from the tumor and identified as having normal germline cell maturation based on a thorough histological analysis. For each seminoma patient, preoperative semen cryopreservation excluded NOA in all cases[9].

Previously collected formalin-fixed paraffin-embedded testis parenchyma, and venous blood from iNOA men was used for retrospective analysis and as the validation cohort. Peripheral blood from a cohort of men with proven fertility and normozoospermia ($n = 41$) was used as a reference cohort for the hormonal profile.

**Ethical approval**. Data collection followed the principles outlined in the Declaration of Helsinki; all patients signed an informed consent agreeing to supply their own anonymous information and tissue specimens. The study was approved by the Institutional Review Board (Ethics Committee IRCCS Ospedale San Raffaele, Milan, Italy), and was composed of 2 protocols for collecting data and using bio-specimens from infertile (Authorization Protocol Infertilità-2015, amended on March 2016) and fertile men (Authorization Protocol URIMALES-2016, further amended on March 2018), and 1 protocol for biobanking (Authorization Protocol URI001-2010, further amended on December 2015). All methods were carried out in accordance with the approved guidelines.

**Testis parenchyma dissociation**. Fresh testis parenchyma was washed twice in PBS and subjected to enzymatic digestion (collagenase from Clostridium histolyticum type I, Merck KGaA; collagenase from Clostridium histolyticum type IV, Merck KGaA; hyaluronidase from bovine testis, Merck KGaA; trypsin from bovine pancreas, Merck KGaA; Deoxyribonuclease I from porcine pancreas, Merck KGaA; all enzymes at 1 mg/ml in DMEM-F12 without serum) for 10 min at 37 °C with gentle agitation, repeated twice (1 ml of digestion medium each time). The supernatants were unified and the cell suspension was filtered through strainers with mesh size 70 μm. The cells were pelleted down by centrifugation (450 g for 5 min), washed twice with 0.4% BSA/PBS, and filtered through a 40 μm cell strainer (Flowmi™ Tip strainers, Merck KGaA) to obtain a single cell. Cell numbers were counted in a Burker chamber and cell viability was evaluated by Trypan blue staining. Cells were suspended in 0.4% BSA/PBS at the concentration of 1000 cells/μl, ready for single-cell sequencing. The recovered cell number was in the range of $3-5 \times 10^6$ cells, with viability > 75%.

**Hormonal profile**. A venous blood sample was drawn from each patient between 7 AM and 11 AM at the time of surgery, after an overnight fast; hormones were measured as previously detailed[32]. Circulating levels of estradiol, tT, and LH were measured in venous blood samples drawn from each patient[32].

**Detection of chemokines**. The chemokines CCL4 and CCL5 in plasma from peripheral blood of iNOA patients and controls were evaluated by Bio-Plex Multiplex immunoassays using Luminex magnetic beads (Human cytokine 27-plex, Biorad) according to manufacturer's instructions.

**Anisotropy measurement**. The degree of organization of fibrils (anisotropy) was evaluated as previously reported[83]. Briefly, 3–5 areas from the same histological section were evaluated for each donor, using 10× magnification acquired with Zeiss AxioImager M2M. Imagines from the histological section were acquired on brightfield to show tissue location, and on darkfield/polarized light to obtain collagen birefringence. Anisotropy was estimated by using the function "measure" and the plug-in FibrilTool[84] in the Image J software. Anisotropy was measured on the basal membrane of the seminiferous tubules.

**Single cell-RNA sequencing performance, library preparation, sequencing, and alignment**. For each experiment, 2500 single cells were processed on the Chromium platform (10×) using the Chromium Single Cell 3′ Library & Gel Bead Kit v3 kit (10×). After quality controls and quantification on TapeStation instrument (Agilent), libraries were sequenced on NextSeq500 or NovaSeq6000 platforms (Illumina) generating around 100'000 reads/cell. Raw sequencing data were demultiplexed with the mkfastq application (Cell Ranger v.3.0.2). UMI-Tools (v.1.0.0) whitelist and extract commands were used to identify and select the number of cell barcodes to use in downstream analysis, setting the cell number to 2000 for donor 1 and donor 2, 5000 for donor 3 and 1500 for donor OA (with the −set-cell-number option). Reads were mapped to the reference genome using STAR v.2.5.3a and assigned to genes with featureCounts v.1.6.4. GRCh38 was used as the

reference genome. Bam files were sorted with samtools (v1.9). Finally, Umi-Tools count was used to processing the UMIs aligned to each gene in each cell to find the number of distinct, error-corrected UMIs mapping to each gene. The UMI count tables of each cellular barcode were used for further analysis.

**Analysis of iGCA samples**. Cell type identification and clustering analysis were performed with the Seurat program (v.3.1.5), within the R environment (v3.6.1). Seurat objects were built from the UMI count tables from each individual experiment. We reported an average of 80,000 reads in each cell, and a median of 1560.0 genes per cell. We filtered out cells with less than 500 genes expressed and with more than 20% reads mapped to the mitochondrial genome (percent.mt). The retained cells were normalized and scaled, regressing out for number of RNA Features, percent.mt and cell cycle. PCA and unsupervised clustering were performed using the top 2000 highly variable genes and the first 20 PCs. Analyzing the resulting clusters in one of the replicas (ID 204—Donor1) we decided to further remove the cells forming one of the clusters presenting more than three mitochondrial genes among its marker genes and a low number of RNA features (indicated as a low-quality cluster in Supplementary information, Dataset S1—spreadsheet marker genes DONOR1). Cell identity for the three iNOA donors is supplied in Supplementary information, Dataset S33. The Seurat objects from different donors were combined using the standard Seurat v3 integration workflow[85], using the IntegrateData function. PCA analysis and clustering were performed on the integrated dataset using the top 2000 highly variable genes and the first 20 PCs. The clustering was performed with the standard Seurat-v3 graph-based clustering approach, based on the functions FindNeighbours and FindCluster, where we set a resolution of 0.2. Cluster visualization was performed with UMAP dimensional reduction technique implemented in Seurat, which is based on uwot R package. Endothelial cells, due to their low amount in the iNOA samples, were not initially identified as a separate cluster. Therefore, we identified endothelial cells from the expression of endothelial marker genes and manually selecting the cluster with the Seurat function CellSelector.

**Analysis of somatic cells from iGCA testis and testis with normal spermatogenesis**. The count matrices for healthy men were obtained from Guo et al. (GSE112013) and Sohni et al. (GSE124263). Annotation differences between these two datasets and our annotation (GENCODE v31 basic annotation) were corrected by updating the gene names from Guo et al. and Sohni et al. to their most recent gene alias in GENCODE v31. This conversion was performed using the Ensembl IDs, which remain comparable across annotation versions. The data were then re-analyzed according to our pipeline, as described above. Guo et al. dataset was also used as a reference to characterize the cell identity of the OA and iNOA samples (as shown in Fig. 1, panels E and F) using the TransferData Seurat function, which projects the PCA structure of a reference onto a query object returning a predicted cell type annotation. Using this approach over 85% of the cell were labeled correctly.

The protocol, then, identified the somatic cells of the healthy donors and integrate those with cells from iNOA men with negative sperm retrieval at microTESE using the standard Seurat v3 integration workflow, with the same parameters described above. After the standard integration workflow, we noticed that 16 cells from the healthy donor datasets clustered with the T-cell cluster from iNOA patients. However, after looking at the expression level of significant immune cell markers (*HLA-DRA*, *CD74*, *HLA-DPA1* and *CD3E*, *TRAC*, *GZMK* for macrophage and T-cell, respectively) in the immune cell clusters (Supplementary information, Fig. S10c), we reassigned those 16 cells to the Macrophage cluster.

After cell type identification, the differential gene expression (DGE) in each cell cluster from the testis of iNOA men with negative sperm retrieval and of men with normal spermatogenesis was performed with the FindMarkers function in Seurat, using default values (Wilcox test, logfc.threshold = 0.25 and min.pct = 0.1). We report all the resulting differential expressed genes in Supplementary Dataset S29.

**Enrichment analysis**. We performed enrichment analysis on the marker genes of each cluster in integrated iNOA samples and on differentially expressed genes between iNOA and CTL samples. We considered a log fold change >0.25 and < −0.25 to unveil both up and down modulated pathways, including only genes with adjusted $P$ value < 0.05. We perform the enrichment analysis with the EnrichR package (v.2.1), which provides an R interface to the 'Enrichr' databases and statistics[86]. In the paper, we mainly described results obtained in the Reactome_2016 datasets and GO_Biological_Process_2018. However, in SI (Supplementary information, Dataset S13–19, Dataset S21–27) we provided a list of all the resulting pathways in GO_Biological_Process_2018, GO_Cellular_Component_2018, GO_Molecular_Function_2018, Reactome_2016, KEGG_2016, WikiPathways_2016, BioCarta_2016, BioPlanet_2019.

To reduce redundancy between genomics pathways and hence aid data interpretation, all the pathways of the Reactome_2016 and GO_Biological_Process_2018 datasets with adj. $P$ value < 0.05 were used to plot Jaccard distance heatmaps for each cell type (Supplementary information, Figs. S4 and S5). Jaccard indices provide a quantitative description of the similarity between two pathways. They were evaluated with the CRAN package *philentropy* (v.0.5.0) and used to compute an adjacency matrix for all possible

pairs of significant pathways, which was then represented with the CRAN package pheatmap (v.1.0.12).

**Cell–cell interactions**. The number of total interactions between cell types was assessed with CellPhoneDB v2.0[28] using the statistical_analysis method, performing 100 iterations for the statistical analysis. We use as input-count-matrix the raw data extracted from the integrated Seurat object (RNA assay) and normalized as prescribed by CellPhoneDB authors (https://www.cellphonedb.org/faq-and-troubleshooting) for iGCA and control samples, separately. The metadata was produced from the Seurat metadata considering the defined cell type assigned after cell clustering. The interaction heatmaps were produced with the CellPhoneDB heatmap_plot function, which uses the pheatmap R library (v.1.0.12).

**LEY cell development signatures**. We combined data from 3 datasets reporting single-cell RNA seq of neonatal, prepubertal, and adult human testes by Cairns group (neonatal: GSE120506; prepubertal: GSE134144, adult: GSE112013 (donor3, 25 yo)) and analyzed those according to the standard Seurat workflow (Supplementary information, Fig. S9a). Then, we subset and re-clustered only the Leydig and Myoid cells, identifying four clusters, one of MYD cells, and three LEY cells at different stages of development (Supplementary information, Fig. S9b). These three clusters were defined as Stage A, associated with neonatal and early prepubertal testis cells (1 yo and 7 yo), Stage B: prepubertal testis cells (13 yo), and Stage C: adult testis cells (14 yo and 25 yo). By DGE analysis, we identified the three stages' marker genes (Supplementary information, Fig. S9c, d and Dataset S30) and defined three signatures considering each cluster's top 20 most significant marker genes. Genes typically associated with spermatogenesis (i.e., PRM1, PRM2, and TNP1), due to ambient RNA in adult testis, were removed from Stage C's gene signature. Furthermore, we identified the somatic cells of the healthy donors from multiple sources (GSE120506, GSE124263, GSE112013, and GSE134144) and integrate those with cells from iNOA men with negative sperm retrieval at microTESE using the standard Seurat v3 integration workflow, with the same parameters described above, in order to compare the expression of Leydig cell immaturity markers such as DLK1 and INSL3 (Fig. 4).

**T-cell characterization**. The T-cells atlas by Szabo et al. (GSE126030), built on 16 samples of CD3+ T cells isolated from tissues and blood, was used as a reference to characterize iGCA T cells (as shown in Fig. 2f) using the TransferData Seurat function, which projects the PCA structure of a reference onto a query object returning a predicted cell-type and tissue annotation.

**LMNA isoforms**. For LMNA differential isoform quantification analysis (Supplementary Dataset S34), kallisto pseudo-alignment tool was used on single-cell reads coming from identified somatic cellular barcodes from both control and iGCA samples. Gencode v37 basic annotation transcript fasta was used as annotation and is reported in Supplementary Dataset S34.

**Immunostaining**. Immunostaining was performed on 5 µm dehydrated FFPE tissue sections after antigen retrieval. Each antibody was titrated on EDTA and citrate-based antigen retrieval, performed by boiling tissue sections at 95 °C for 30 min in EDTA Decloaker pH 8.5 (Biocare Medical, Pacheco, CA; cod. CB917) or Reveal Decloaker pH 6.0 (Biocare Medical; cod. RV1000M), respectively. Next, tissue sections were allowed to cool and then kept at room temperature for 20 min. Tissue sections were washed twice in PBS and permeabilized in PBS containing 0.4% Triton (Merck Life Science, Germany, cod T8787) and 0.1% donkey serum (Abcam, United Kingdom, cod. ab7475) for 20 min at room temperature; the aspecific sites were blocked with blocking buffer (10% donkey serum in PBS-Triton 0.4%) for 1 h at room temperature. Tissue sections were incubated with primary antibodies in the humid box at 4 °C for 16 h. After two washes in washing buffer (PBS containing Triton 0.4%), secondary fluorescence-conjugated antibodies were added in a dark humid box and incubated for 1 h at room temperature. Slides were washed twice with washing buffer and nuclei were stained with NucBlue Live ReadyProbe Reagent diluted 1:10 in PBS for 30 min at room temperature. After removing the excess of this solution, tissue sections were rinsed in PBS and mounted with Fluorescence Mounting Medium (Agilent, USA, cod. S302380-2). Images were acquired by using Olympus FluoVIEW FV3000RS Confocal, objective UPLXAPO 60XO (NA 1.42) Oil, and imaging analysis performed with the software ImageJ. Primary and secondary antibodies, including dilution of each antibody and the antigen retrieval used for each antibody, are reported in the Supplementary Dataset S35.

**Testis datasets used in this study: similarities and differences**. Supplementary Dataset S36 lists all the datasets used in this study, including our own with data from 3 iGCA and 1 OA testis. Details show (i) the age of the donors, (ii) sample processing and sequencing, and (iii) how the data were used in our study.

In our and other studies, similar procedures had been used for tissue dissociation, consisting of enzymatic digestion completed in 30 min. The cell processing was also similar, and the RNA-seq was performed with the 10× Genomic system on the Chromium platform. Collection and storage of the tissues were the main differences between our study and the others we used. In our study, we used fresh testicular tissue obtained during microTESE intervention, and the tissues were kept in cold Krebs solution and processed within 15 min from the collection in the surgery room (as reported above). In the studies from others, the authors used tissues from autoptic donors[15] or collected after vasectomy[24]. The tissue specimens were frozen and then thawed for RNAseq.

To overcome the potential bias in the DGE and modulated pathways analyses of comparing fresh vs. autoptic or frozen tissues, we have firstly combined control donors from multiple input datasets (from Cairns' and Wilkinson's groups). Secondly, we have used the integration method available in Seurat v3, which is aimed at correcting for technical differences between datasets (i.e., batch effect correction), and performing comparative scRNA-seq analysis across different experimental conditions. Lastly, we added a comparative analysis of iNOA samples with the RNAseq dataset obtained from the testis of an OA man with normal spermatogenesis processed in our lab. We found similar DEG and deregulated pathways in LEY and MYD cells as in the comparison with others' datasets (Supplementary Fig. S6).

In conclusion, we are confident that the comparisons made in this study are not biased by the tissue collection and storage methodologies.

**Statistical analyses**. Continuous variables were expressed as medians (IQR), and significance was analyzed by two-tailed Mann–Whitney or Kruskal–Wallis test, with $P < 0.05$ considered statistically significant.

**Reporting summary**. Further information on research design is available in the Nature Research Reporting Summary linked to this article.

## Data availability

Library construction was performed through the Chromium Single Cell 3′ v3 Reagent kit &Chip Kits (10x Genomics). The single-cell RNA sequencing data generated in this study (one testis with obstructive azoospermia and three testes with iGCA) have been deposited in the GEO database under accession code "GSE154535". The testicular cells scRNA-seq data from human adults with normal spermatogenesis are publicly available in the GEO database from the study of Guo and colleagues[15] under accession codes "GSE112013", "GSE120506", and "GSE134144", and Shomi and colleagues under accession code "GSE124263". Data for the T cell atlas are publicly available from in the GEO database, under accession code "GSE126030". All other relevant data supporting the key findings of this study are available within the article and its Supplementary Information files or from the corresponding author upon reasonable request. Source data are provided with this paper.

## Code availability

All codes associated with this manuscript have been uploaded to GitHub https://github.com/volpesofi/iGCA_scRNAseq_analysis[87] and are available in Zenodo at the following https://doi.org/10.5281/zenodo.5184876.

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

## Acknowledgements

We thank Dr. Giulia Villa, Dr. Federico Belladelli, Dr. Andrea Baudo, and Dr. Edoardo Pozzi (University Vita-Salute San Raffaele, Milan, Italy) for coordinating the human testicular tissue collection; Dr. Marco Morelli and Dr. Dejan Lazarevic (IRCCS Ospedale San Raffaele, Milan, Italy) for their support with biostatistical analyses and single cell sequencing, respectively; Dr. Cesare Covino from ALEMBIC, an advanced microscopy laboratory established by IRCCS Ospedale San Raffaele and Università Vita-Salute San Raffaele. This work was supported by URI-Urological Research Institute free funds; the funding source did not have any role in the present study.

## Author contributions

Conceptualization: M.A.; Methodology: M.A. and A.S.T.; Bioinformatics analysis: A.S.T. and J.M.G-M.; Formal analysis: M.A. and A.S.T.; Investigation: M.A., F.P., A.S., A.S.T., I.L., M.N., G.A., S.G., A.A., and F.G.; Resources: A.S.; Data curation: M.A. and A.S.T.; Writing-original draft: M.A. and A.S.; Writing—review and editing: M.A., A.S.T., F.P., J.M.G.-M., G.T., S.G., A.A., A.S. and F.M.; Funding acquisition: F.M. and A.S.; Supervision: M.A. and A.S.

## Competing interests

The authors declare no competing interests.

## Additional information

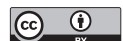

