## [Peer Review File · Nature Communications]

Reviewers' Comments:

Reviewer #1:

Remarks to the Author:

Comments:

This paper reports the comparison of human idiopathic germ cell aplasia and normal spermatogenesis donor single-cell transcriptome. Through analysis, the author found that somatic cells have early aging processes. The above conclusions are mainly based on the following observations: detection of H2AX in Sertoli cells; up-regulation of the expression of some cytokines, which are considered to be senescence-related secretory phenotype (SASP). Although some discoveries appear impressive, following the key claims through their supporting evidence repeatedly shows examples of little support. The paper proposes that the transformative findings of the research are not sufficient, and the experimental evidence related to aging needs to be strengthened. In the end, I am not convinced that the authors have made the connection between germ cell aplasia with aging events. The current version is obviously not suitable for publication in Nature Communications.

Major comments:

1. The three samples in the article have a large difference in the number of cells obtained. What is the difference between the three samples? Morphologically, the diagnosis of iNOA needs to be combined with clinical information, and it is also necessary to compare the differences between the three samples. Although these samples are called iGCA clinically, the actual differences may be very large.
2. In 2018, single-cell sequencing published by Wang et al. has compared the difference between iNOA and normal, and reached a conclusion similar to this article. It also found that the H2AX phosphorylation level of Sertoli cells is increased. Conceptually, the novelty of this article is not strong. Does the author also need to compare the data in the article with the conclusion of this article? There is no relevant reference in this article.
3. The immaturity of Leydig cells is an interesting observation, but only the transcript changes of DLK1 and Notch2, the evidence is insufficient, and the corresponding evidence needs to be given from the protein level or function.
4. For the conclusion of aging, the author only has transcript information and DNA damage-related staining, whether the classic aging-related detection can be performed on related tissues. In addition, in Leydig cells, the author's staining of gamma H2AX is mainly in the cytoplasm. This means that DNA damage is serious? If such staining appears in the cytoplasm, co-staining related to cell death is required.

Specific comments:

1. P5-93: Author claim to have identified two new cell types, T-cells and fibroblasts, it is an interesting and important finding, however, no other evidence were showed in this study such as FACS, IHC or ICC.
2. Figure 5B-C: Since the repeatability of the three samples is poor, the author needs to provide the clinical information and pathological section information of all the three people, etc., to explain whether the difference is due to the sample itself or the experiment process.
3. P7-128: The data analysis of CFTR mutations sample is not relevant to the subsequent results and conclusions of this article. This part may remove from this study?
4. P7-129: "To enable comparison and exclude batch effects, the datasets from the cells obtained in our 3 iGCA testes were integrated with those from the three testicular samples with normal spermatogenesis from Guo and colleagues". As describe in P21-424, human testicular samples for scRNA-seq was performed with fresh testicular tissue in this study, but in Guo's study, all samples underwent cryopreservation and thawing. Authors should combine data from other laboratory or other evidence to prove the difference between iGCA and normal tissues (especially energy metabolism and cell apoptosis) were does not come from the way the samples were handled.
5. P8-141-145: "Interferon gamma mediated signaling" and "Type I Interferon signaling" were

enriched in MYD and LEY, does these two cells showed some interaction with resident immune cells?

6. Figure 6B: the IHC figures were not clearly enough, whether immunofluorescence staining can better explain this result.

7. P14-302: Why low levels of tT associated to an immature status of LEY but also a marker of risk of early aging in iNOA men, the relate reference papers should be cited here.

8. P16-316: the ranscriptional atlas of the testicular somatic cell populations associated with human iGCA have been reported by Wang, et al in 2018. Their work should be cited here.

9. P16-329: The increased deposition of collagen in the basal membrane of the iGCA patients was not a new finding, some previous studies have reported similar results in a larger population (such as Liu Y, et al. Fertil Steril. 2014 Jul;102(1):54-60.e2.), these studies should be discussed and cited.

Reviewer #2:

Remarks to the Author:

The manuscript submitted by Alfano et al. is entitled 'Human idiopathic germ cell aplasia is characterized by aged and inflamed testis somatic components with an active DNA Damage Response. The aim of this study was to unveil the occurrence of dis-regulated transcriptional pathways in testicular somatic cell populations in patients with idiopathic germ cell aplasia (iGCA). The authors make a number of big claims, two of which are: That the altered transcriptional pathways in testicular somatic cell types of iGCA samples are fostering early ageing processes in all somatic cell types. Moreover, the authors claim that Leydig cells display an immature profile and that altered expression of parentally imprinted genes was found in myoid and immature Leydig cells, which the authors suggest are involved in the hyalinization of the basal membrane of the seminiferous tubules. Finally, authors claim that this study not only helps to understand the pathology of iGCA but also 'the associated clinical manifestations of early aging process and related oncological and non-oncological diseases'.

Better characterization and understanding of the mechanisms leading to germ cell aplasia are certainly desirable and would be highly interesting to the community of reproductive medicine/biology. However, in its current form, the conclusions are not supported by the datasets and the overall study design is difficult to discern. Moreover, the authors rather chose to use the datasets to address commonly known alterations of the testicular tissues associated with infertility (i.e. thickening of the basement membrane (Dobashi et al. 2003; Volkmann et al., 2011) and then use the scRNA-Seq dataset to supporting respective alterations, rather than developing new concepts with the available single cell data. Thereby, this study remains below its actual potential. Another concern is the over interpretation of results, which is outlined as one example for the Leydig cells below:

- The authors analysed testicular somatic cells from 3 patient with iGCA, lacking the germ cell population (SCO-phenotype) and compared results to normal data obtained from a sample with full spermatogenesis and published datasets. By analysis of SCO samples the authors indeed generated interesting datasets of testicular somatic cells, which are rarely represented in published datasets of testicular tissues with full spermatogenesis, in which the germ cells account for about 75% of the total cell population. However, a balanced number of normal and iGCA samples would have been desirable.

- For Leydig cells, authors conclude that they display an immature phenotype based on the 'overexpression' of the DLK1 gene. To prove the immaturity of Leydig cells compared to the normal situation, authors should have provided analyses also at protein level, comparing the two patient groups. Apart from that, they could have employed published data (Guo et al. 2019) for comparison with immature human Leydig cells obtained from prepubertal boys.

- As only transcriptional data is provided, the interpretation that 'Leydig cell immaturity, at least since puberty, may favour the boost of a chronic pro-inflammatory environment, which, on the one hand is responsible for the fibrosis of the seminiferous tubules and on the other eventually for germ cell depletion' is one example of an over interpretation of results. As the authors do not have any information whether germ cells were ever present in the testicular tissues of this patient group, the suggestion that germ cells were depleted is pure speculation. It would be important to compare the datasets with the available data from immature human testicular tissues to elucidate how the situation in these testicular tissues compares to normal development.

In general, this study would greatly benefit from a more focused and in depth analyses of selected issues that are touched by this manuscript (facilitating essential validation and comparison to available datasets) and importantly an interpretation that is based on the datasets.

Reviewer #3:

Remarks to the Author:

In this manuscript the authors perform a detailed single cell RNA-Seq analysis of testis samples from men with idiopathic germ cell aplasia (iGCA) and one sample from a case of obstructive azoospermia (OA) that retains germ cells. Multiple different somatic cell types including immune cells were identified and transcriptional profiles compared to a previously published reference dataset from normal human testis. Differentially expressed genes in some of the somatic cell types of iGCA samples including inflammatory markers and genes associated with extracellular matrix. Based on gene expression analysis between the different cell types, many cell pathways were downregulated in most somatic cell types of iGCA samples including those associated with protein synthesis. Transcriptional profiles also suggested that hormone producing Leydig cells were developmental immature and this correlated with reduced testosterone levels in the circulation of infertile men with non-obstructive azoospermia (NOA). Analysis of testis sections from cohorts of NOA men confirmed changes in extracellular matrix deposition and suggested a chronic inflammatory response. Systemic levels of pro-inflammatory cytokines were also elevated in NOA patients. Combined, the authors propose that iNOA testis may develop as a result of chronic testicular (and systemic) inflammation and a premature ageing process within the testis somatic cells.

The manuscript will be of interest to the field and single cell datasets generated from iNOA samples invaluable. Mechanisms of most NOA cases are unclear. In general, the analysis has been performed in a careful and thorough manner and the identification of systemic indicators of iNOA very interesting. However, some points require further exploration and there are major concerns over the control samples used for the analysis.

Major points:

1. The authors subject samples from iNOA and OA individuals to single cell RNA-Seq but no controls (fertile, age-matched men) were sampled and analysed in their manuscript. To assess changes in gene expression patterns in testis somatic cells from patients, they compare to samples previously analysed by the group of Bradley Cairns. However, more consideration needs to be given as to whether these are suitable as controls. Are they age matched? Was cell extraction and processing equivalent? Concerns as to the suitability of the control data used is illustrated by the fact that many common cell pathways (48) are predicted to be downregulated in all different somatic cell types present in the iNOA testis, including many ribosomal protein genes. This seems very unlikely given the diversity of cell types within the testis and suggests that the controls are not suitable. The iNOA samples should also be compared to control datasets from other labs e.g. Miles Wilkinson. In addition, although the OA sample may not be ideal as a control, at least it was processed alongside the iNOA samples so a comparison between OA and iNOA samples would be very informative. Are similar pathways misregulated in iNOA somatic cells versus those from the OA patient?

2. T cells and fibroblasts are identified from iNOA testis but not present in the control single cell datasets used. It is unclear if this is because T cells and fibroblasts are only present in iNOA testis or they are just not represented in the Cairns dataset. Is this true for other available human testis single cell datasets? This limits the conclusions that can be drawn about these populations. Are there alternative single cell datasets of T cells and fibroblasts from other human tissues that can be used as comparison? For example, are there unique features of T cells present in iNOA testis versus other tissue-resident T cells?

3. Based on single cell analysis, Leydig cells are proposed to be functionally immature in iNOA samples, which correlated with reduced testosterone production. INSL3 is a maker of Leydig cell maturity and was analysed in iNOA Leydig cells. Confusingly, more cells were found to be INSL3

positive in iNOA samples than those from controls although of the cells that were positive in controls, the median expression level of INSL3 was higher than that of iNOA samples. Does this indicate that the control samples contained many more immature Leydig cells than the iNOA samples (unexpected) but that those cells which were mature were “more mature” than those from iNOA samples? This point needs further exploration and doesn’t seem consistent with the broad conclusion that iNOA Leydig cells are more immature than controls. How do the iNOA Leydig cells compare to cells from pre-pubertal immature human testis?

4. Changes in expression of imprinted genes are found in iNOA somatic populations although the significance not explored. Are epigenetic regulators known to be involved in maintaining gene imprinting mis-expressed in iNOA cells? What is the potential functional significance of this observation?

5. Histological analysis of iNOA testis indicated dynamic changes in collagen I (increased) and collagen IV (decreased). However, the single cell analysis seemed to show only increased expression of collagen subunits in somatic cells. How are these two observations reconciled?

6. Changes in the localisation pattern of HMGB1 are suggested to indicate cellular senescence and chronic inflammation in somatic cells of iNOA testis. Are other known indicators of cellular senescence (e.g. cell cycle-related proteins) and inflammatory signalling changed in iNOA samples? On a related note, the gH2A.X staining suggested DNA damage in iNOA cells but no signal was apparent in normal testis. Typically, meiotic germ cells should be gH2AX positive due to the DNA breaks specifically induced during that process. Are germ cells positive in the control samples? Are known components/targets of the DNA damage response differentially expressed in iNOA cells?

7. Changes in systemic levels of vitamin D are proposed to indicate enhanced ageing in iNOA individuals. Why have the authors focused specifically on vitamin D? Is this routinely used as a marker of the aging process in humans? Additional details on the utility of this marker should be provided.

8. A major issue concerns the over-arching conclusion of the study – that chronic inflammation and other changes in testis somatic cells (senescence etc) are responsible for loss of germ cells in iNOA patients. However, as samples are not analysed at earlier stages during disease progression, can it be excluded that the observed changes in somatic cells are a result rather than a cause of the loss of germ cells at various stages of testis development/maturation? In essence, without functional data supporting a causative role for the described changes in somatic cells in germ cell aplasia, this conclusion cannot be made. The discussion needs to be modified accordingly.

Minor points

1. Lines 340-341 – to exclude the possibility that endothelial cells play a role in germ cell aplasia seems premature given that they are proposed play important roles in rodent germ cell maintenance and they do have changes in gene expression versus control. Functional data would be required to exclude this possibility.

2. Lines 375-377 – effects of inflammatory cytokines on stem cell function are discussed. However, the quoted references concern hematopoietic and intestinal stem cells. Any evidence that they have a similar effect on germline stem cells?

Reviewer #1 (Remarks to the Author):**Comments:**

This paper reports the comparison of human idiopathic germ cell aplasia and normal spermatogenesis donor single-cell transcriptome. Through analysis, the author found that somatic cells have early aging processes. The above conclusions are mainly based on the following observations: detection of H2AX in Sertoli cells; up-regulation of the expression of some cytokines, which are considered to be senescence-related secretory phenotype (SASP). Although some discoveries appear impressive, following the key claims through their supporting evidence repeatedly shows examples of little support. The paper proposes that the transformative findings of the research are not sufficient, and the experimental evidence related to aging needs to be strengthened. In the end, I am not convinced that the authors have made the connection between germ cell aplasia with aging events. The current version is obviously not suitable for publication in Nature Communications.

Major comments:

1. The three samples in the article have a large difference in the number of cells obtained. What is the difference between the three samples? Morphologically, the diagnosis of iNOA needs to be combined with clinical information, and it is also necessary to compare the differences between the three samples. Although these samples are called iGCA clinically, the actual differences may be very large.

We agree with this Reviewer that clinical information is relevant. The Methods section has been amended by including clinical descriptions (page 32, paragraph "MicroTESE, sperm retrieval and tissue processing")

The dissociation procedure used to isolate single cells from the testis parenchyma was validated in our lab using non-neoplastic tissues from orchiectomies and is superimposable to that used by other authors. The recovery from the dissociation procedure of our sample is >90% and with >80% cell viability. Differences in cell yield from the 4 patients from our lab were likely due to the amount of tissue removed during the microTESE interventions.

2. In 2018, single-cell sequencing published by Wang et al. has compared the difference between iNOA and normal, and reached a conclusion similar to this article. It also found that the H2AX phosphorylation level of Sertoli cells is increased. Conceptually, the novelty of this article is not strong. Does the author also need to compare the data in the article with the conclusion of this article? There is no relevant reference in this article.

We thank the Reviewer for highlighting this piece of information from Wang et al.¹

Wang et al. reported gH2AX staining in meiotic spermatocytes from control and OA testes and the accumulation of gH2AX signal in NOA, but not OA, Sertoli cells. In addition, Wang et al also identified the upregulated pathway "Generation of precursor metabolisms and energy" by analysing the scRNA-seq transcriptome profiles of 174 testicular cells from one NOA donor (24 years old).

As requested in Point 4, we have repeated gH2AX analyses using the immunofluorescence protocol. The new analysis allowed us to appreciate accumulation of gH2AX in Leydig and Myoid cells of iGCA, but not in the Sertoli cells, as reported in our previous version of the manuscript. The gH2AX in the meiotic spermatocytes of control tissues suggests the fidelity of our staining. We are sorry for having previously shown incomplete information.

We amended the Results section (Figure 6B) and cited the work by Wang et al. accordingly. We also amended the Discussion section by citing the work by Zhao et al, which is in agreement with the immature state of the Sertoli cells in iNOA men.

3. The immaturity of leydig cells is an interesting observation, but only the transcript changes of DLK1 and Notch2, the evidence is insufficient, and the corresponding evidence needs to be given from the protein level or function.

We agree with the Reviewer that the immaturity of the Leydig cells plays a central role in the NOA.

In the original version of the manuscript, we have inferred the immature state of Leydig cells in iNOA men by the DE transcription profiles of DLK1/NOTCH2, IGF2, INSL3, and HSD17B3. At systemic level, we showed decreased Testosterone in blood from iNOA vs age-matched fertile individuals.

In the revised version of our study, we deepened the investigation on the immature state of Leydig cells at transcriptional level by comparing our dataset vs those reported by Guo and Sohni, which included data from neonatal and prepuberal testis²⁻⁴. We now report that iGCA Leydig cells are stuck at an immature stage, with a transcriptional signature recapitulating the stage A-B of maturation identified in neonatal-prepuberal human testis (as shown in Figure 4 and in the Supplementary Information, Figure S20). In addition to decreased peripheral level of testosterone in iNOA men, the iGCA Leydig cells immature state was validated at protein level by showing a reduced staining for INSL3 protein in the Leydig of iGCA vs control testis (Figure 4C). In the Supplementary Information (Figure S17-S19), additional staining shows the over-expression of DLK1 and the reduced expression of IGF2 to further validate LEY immaturity.

4. For the conclusion of aging, the author only has transcript information and DNA damage-related staining, whether the classic aging-related detection can be performed on related tissues. In addition, in leydig cells, the author's staining of gamma H2AX is mainly in the cytoplasm. This means that DNA damage is serious? If such staining appears in the cytoplasm, co-staining related to cell death is required.

We thank the reviewer for raising the important issue of senescence.

In the revised manuscript, beside the cytoplasmic localization of the SASP HMGB1, the comparison with GenAge database (<https://genomics.senescence.info/legal.html>), and the higher level of CCL4 and CCL5 chemokine transcripts and blood chemokine levels, we now show the accumulation of p16 protein in Leydig cells.

In this revised version of the study, we have also added new finding on misshapen nuclei and alteration in the ratio of Lamin A and C, which well fit with the senescent phenotype of Sertoli and Leydig cells.

We agree with this Reviewer that some of our conclusions were not completely supported by data. Therefore, conclusions have been toned down by referring the premature aging only to the testis tissue. We have removed any sentence that might suggest a general early ageing process in the iNOA men.

As for the staining for gH2AX in the cytoplasm of LEY in the iGCA testis, we amended the manuscript according to the reply to question #2 of the same Reviewer.

Specific comments:

1. P5-93: Author claim to have identified two new cell types, T-cells and fibroblasts, it is an interesting and important finding, however, no other evidence were showed in this study such as FACS, IHC or ICC.

We thank the Reviewer for highlighting this point, and we further analysed our data.

Regarding the characterization of the STRO cluster, we took the specific makers of this cluster (FDR < 0.05 & log2FC >1, Table S11) and compared their expression with Immgen dataset^{5,6}, containing hematopoietic as well as non-hematopoietic stromal cells. As seen in Reviewer Fig1, fibroblast-reticular cell type (FRC) expressed the highest levels of the STRO markers. Further analyses detected that the stromal cell cluster expressed markers of pericytes, more specifically the RGS5⁷⁻⁹ and TPM2⁹ genes.

Pericytes are present in several tissues in the body, testis included, but had never been described transcriptomically in the human testis. Indeed, in mice, peritubular myoid cells (MYD), vascular smooth muscle and pericytes have been recently shown to derive from Nestin-positive progenitor cells, and this origin might explain why pericytes express markers common to smooth muscle cells^{10,11}. These Nestin-positive progenitors have been proposed as a potential source for Adult Leydig cells (LEY) and even for potential treatment of Leydig cell dysfunction¹². Although we could speculate that the presence of a detectable STRO (perivascular) cell in iNOA patients might represent an attempt to sustain the differentiation in LEY that are dysfunctional, we are not confident that this information is validated in the present study.

We have now amended our manuscript, adding the more precise identification and the references used for the characterization of the STRO cells (pericytes).

Reviewer Figure 1 Heatmap of expression of STRO cluster specific markers ($\text{avg_logFC} > 1$, $\text{adj.P.val} < 0.05$, in Supplementary Table S0-S11) within the Immgen dataset^{5,6} pointing to stromal cells (Fibroblastic-Reticular cells).

Regarding T lymphocytes, IHCs on FFPE testis was not able to assess with certainty the specificity of staining, due to scarcity of TCL (98 cells only were detected by scRNAseq) in the 3 testes parenchyma. On the other hand, it was not possible to perform FACS analysis of fresh tissues due to the COVID-19 pandemic; since March 2020, surgery for benign diseases has been stopped, including microTESE surgery that applies to azoospermic men necessary for the recovery of the testicular parenchyma to attempt sperm recovery.

A recent study, which was not published at the time of the first submission, also reports the presence of T-cells in testis from 4 healthy donors (Shami et al.¹³) and the identified T-cells were CD3D, TRAC, TRBC2, and CD69 positive and 23% of those expressed CD8. Shami et al. samples were a pool of testis tissues tissue from a tissue bank with regular spermatogenesis from adults ageing from 20-40 years, therefore age-matched with the iNOA men used in our study.

While it is possible that Guo et al. and Sohni et al. normal testes did not show any resident T cell due to any technical reasons, in all our iGCA samples almost all resident T-cells were CD8 positive and expressed markers of active cytotoxic CD8 T lymphocytes like GZMM and GZMK (Figure 2H).

In order to robustly exclude the blood origin of the resident CD8 T-cells in iGCA, we compared our datasets with a single-cell dataset (GSE126030¹⁴) of T-cells collected in blood, bone marrow, lung and lymph nodes. This analysis confirmed that the vast majority of iGCA T-cells were mapped to CD8 Tissue Resident Memory (TRM) T lymphocytes and retained markers expressed from Lung tissue resident T cells, not-present in circulating T-cells (added to the main text as Figure 2G).

We amended the manuscript showing the new characterisation of TCL and introducing the reference to the study of Shami et al. when describing the presence of a small subset of T-lymphocytes in the normal adult testes. Furthermore, we erased the term novel from our manuscript when referring to the finding of T-cells in the normal testis.

2. Figure 5B-C: Since the repeatability of the three samples is poor, the author needs to provide the clinical information and pathological section information of all the three people, etc., to explain whether the difference is due to the sample itself or the experiment process.

We agree with this Reviewer that clinical information is relevant. The Methods section has been amended by including clinical descriptions (page 32, paragraph "MicroTESE, sperm retrieval and tissue processing")

The dissociation procedure used to isolate single cells from the testis parenchyma was validated in our lab using non-neoplastic tissues from orchiectomies and is superimposable to that used by other authors. The recovery from the dissociation procedure of our sample is >90% and with >80% cell viability. Differences in cell yield from the 4 patients from our lab were likely due to the amount of tissue removed during the microTESE interventions.

3. P7-128: The data analysis of CFTR mutations sample is not relevant to the subsequent results and conclusions of this article. This part may remove from this study?

We regret to disagree with the reviewer. We believe that the comparison of OA/CFTR cells purified in our lab compared to iNOA cells allows the validation of Guo and Sohni datasets as bona fide CTLs.

At the beginning, the tissue and relative scRNA-seq from the OA patient was used to properly setup the experimental procedures, the bioinformatics analyses and cell cluster identification. We still believe that the information from the OA sample is relevant to this study as it is the only control sample that we could get using the same procedure and technology and represents an internal control for the quality of the data obtained in this study.

^{2,32,31,2}We have added an integration analysis of the 3 iNOA samples vs the OA sample as an additional control which is shown in Supplementary Information as Figure S16. The UMAP of Figure S16, shows that most somatic cells from the OA patient mapped to iNOA LEY and MYD. When we compared the transcriptomes, LEY and MYD cells from the iNOA patients vs. the OA patient show upregulation of ECM organization, Scavenging by Class A receptor, Pre-NOTCH transcription and translation, and downregulation of ribosome biogenesis and Selenocysteine synthesis. These results overlap what we obtained in the comparison with Guo and Sohni CTLs ^{2,3}.

4. P7-129: "To enable comparison and exclude batch effects, the datasets from the cells obtained in our 3 iGCA testes were integrated with those from the three testicular samples with normal spermatogenesis from Guo and colleagues". As describe in P21-424, human testicular samples for scRNA-seq was performed with fresh testicular tissue in this study, but in Guo's study, all samples underwent cryopreservation and thawing. Authors should combine data from other laboratory or other evidence to prove the difference between iGCA and normal tissues (especially energy metabolism and cell apoptosis) were does not come from the way the samples were handled.

We thank the reviewer for pointing out the need for considering additional datasets. In the revised version of this manuscript, we added cells from 2 healthy donors from Sohni et al., obtained from adult testicular biopsies from two fertile men aged 37- and 42-years, undergoing vasectomy, that allowed us to reach the same main conclusions as before. Moreover, the comparison of iNOA samples with the OA patient (whose RNA-seq was extracted, processed and sequenced in our laboratory) supports the same conclusion in terms of downregulation of ribosomal genes (related to energy metabolism) (Supplementary information, Figure S16). The results presented from Figures 1G

to Figure 6L and the manuscript have been amended accordingly to include the additional control datasets.

5. P8-141-145: “Interferon gamma mediated signaling” and “Type I Interferon signaling” were enriched in MYD and LEY, does these two cells showed some interaction with resident immune cells?

We thank this reviewer for giving us the opportunity to expand our analyses.

We are aware of the possibility of investigating the putative interaction between different cell types in a tissue through the study of their transcriptome at single cell level. We started to explore this possibility using the recently published tool CellPhoneDB¹⁵. Data analyses predict increased interactions of Leydig and Myoid cells with macrophages in iGCA testis compared to what is found in controls. These results are now shown in Figure 2I and Table S19.

6. Figure 6B: the IHC figures were not clearly enough, whether immunofluorescence staining can better explain this result.

As requested, we have repeated the analysis for gH2AX using the immunofluorescence staining. The new analysis increased the information retrieved, for which we thank the Reviewer. We detected bona fide DNA damage in Leydig and Myoid cells, but not in Sertoli cells as reported in the previous version of the manuscript. We amended the section of Results and the Discussion accordingly

7. P14-302: Why low levels of tT associated to an immature status of LEY but also a marker of risk of early aging in iNOA men, the relate reference papers should be cited here.

The Reviewer is referring to the paragraph at page 14 (lines 297-305).

In particular, we think the Reviewer is referring to the statement “Hence, LH/tT ratio is representative of the idiopathic primary testicular dysfunction but is also suggestive of a phenotype of senescence and might serve as a proxy of higher risk of early ageing in iNOA men”.

In this revised version of our study, we deepened the analyses Leydig cells immaturity, showing that they are blocked at the neonatal-prepuberal stage (Figure 4). It is globally accepted that the LH/Tt ratio is increasing during aging¹⁶. We agree that at present our does not provide a proof for the sentence “LH/tT ratio is suggestive of a phenotype of senescence and might serve as a proxy of higher risk of early ageing in iNOA men”.

We removed the sentence, and now the paragraph closes with the following sentence “Therefore, low levels of tT potentially associated to an immature status of LEY cells, were associated to a compensatory positive feedback in terms of LH synthesis.

8. P16-316: the transcriptional atlas of the testicular somatic cell populations associated with human iGCA have been reported by Wang, et al in 2018. Their work should be cited here.

We thank the Reviewer for highlighting the paper from Wang et al.¹, which has been properly cited in the revised manuscript.

9. P16-329: The increased deposition of collagen in the basal membrane of the iGCA patients was not a new finding, some previous studies have reported similar results in a larger population (such as Liu Y, et al. *Fertil Steril.* 2014 Jul;102(1):54-60.e2.), these studies should be discussed and cited.

*We agree with this Reviewer that hyalinization of the seminiferous tubule was previously reported by several authors in association with germ cell aplasia/SCOS, since the '70s¹⁷. The study mentioned by the Reviewer (Liu Y et al. *Fertil Steril.* 2014 Jul;102(1):54-60.e2.) applied the Raman spectroscopy to discriminate seminiferous tubules with complete vs incomplete spermatogenesis based on the morphological alterations of the testicular lamina propria and performed IHC analysis to validate the information obtained through Raman spectroscopy (original figure from Liu Y. et al. reported below).*

Immunohistochemical staining from the Figure 3 by Yufei Liu et al (*Fertil Steril.* 2014 Jul;102(1):54-60.e2). In OA tubules, the testicular LP was positive for type IV collagen and laminin but negative for type I and type III collagens. The type IV collagen and laminin were detected in the whole layer of LP. In NOA tubules, the type IV collagen and laminin were more intensely stained but differently from OA tubules; the expression of type I and type III collagens was also detected, especially in the basement membrane. In addition, the lumens of NOA tubules were also stained.

We carefully revised the figure from Liu Y. et al., but we could not appreciate the collagen IV deposition in the basal membrane of OA testis that, on the other side, seems overrepresented in the vassal lamina of NOA tubules. Unexpectedly, the IHC stain detected collagen I, collagen III and Laminin inside the seminiferous tubules while the correct localization would be in the basal lamina. Although Raman spectroscopy analysis reached conclusions similar to ours, the same information are questionable from the histochemical point of view.

To the best of our knowledge, our study is the first to clearly show that i) collagen I is over-deposited ii) while collagen IV is decreased in iGCA basal lamina, iii) seminiferous tubules with spermatogenesis and positive for collagen IV in ECM, colocalise in the same parenchyma with seminiferous tubule without spermatogenesis and negative for collagen IV but over positive for collagen I, and iv) collagen organization undergone a visible remodelling (anisotropy, Figure 5F and 5G) in the hyalinized basal membrane of seminiferous tubules associated with germ cell aplasia.

We amended our manuscript citing a review paper published in 2011¹⁸, reporting the previous evidence of the overall hyalinization of the seminiferous tubule in infertile patients.

Reviewer #2 (Remarks to the Author):

The manuscript submitted by Alfano et al. is entitled ‘Human idiopathic germ cell aplasia is characterized by aged and inflamed testis somatic components with an active DNA Damage Response. The aim of this study was to unveil the occurrence of dis-regulated transcriptional pathways in testicular somatic cell populations in patients with idiopathic germ cell aplasia (iGCA). The authors make a number of big claims, two of which are: That the altered transcriptional pathways in testicular somatic cell types of iGCA samples are fostering early ageing processes in all somatic cell types. Moreover, the authors claim that Leydig cells display an immature profile and that altered expression of parentally imprinted genes was found in myoid and immature Leydig cells, which the authors suggest are involved in the hyalinization of the basal membrane of the seminiferous tubules. Finally, authors claim that this study not only helps to understand the pathology of iGCA but also ‘the associated clinical manifestations of early aging process and related oncological and non-oncological diseases’. Better characterization and understanding of the mechanisms leading to germ cell aplasia are certainly desirable and would be highly interesting to the community of reproductive medicine/biology. However, in its current form, the conclusions are not supported by the datasets and the overall study design is difficult to discern. Moreover, the authors rather chose to use the datasets to address commonly known alterations of the testicular tissues associated with infertility (i.e. thickening of the basement membrane (Dobashi et al. 2003; Volkmann et al., 2011) and then use the scRNA-Seq dataset to supporting respective alterations, rather than developing new concepts with the available single cell data. Thereby, this study remains below its actual potential.

We thank the Reviewer for appreciating the potential impact of our study to better understand the idiopathic form of germ cell aplasia. We also thank this Reviewer for highlighting the poorly characterized points, whose removal allowed us to sharpen the focus and novelty of our study in the revised manuscript.

We agree that some of our novel findings were over-interpreted, and in the revised version of the manuscript we erased the speculations about early ageing, including the sentence reported in the Discussion “the associated clinical manifestations of early aging process and related oncological and non-oncological diseases”. In this revised version of our study, we carefully supported the immature state of Leydig cell, showing that they are blocked at the neonatal-prepubertal stage (Figure 4). Furthermore, we also validated at protein level some features obtained by transcriptome analysis, such as the reduced level of INSL3 expression in the Leydig of iGCA vs. control testis. In the Supplementary Information we also show the over-expression of DLK1 and the reduced expression of IGF2 at protein level in iGCA vs. control adult testes (Figure S17-S19).

Regarding the comment on the “use the datasets to address commonly known alterations of the testicular tissues associated with infertility (i.e. thickening of the basement membrane (Dobashi et al. 2003; Volkmann et al., 2011)”: we agree with the Reviewer that that seminiferous tubule hyalinization was previously reported in association with germ cell aplasia/SCOS, by several authors since the ‘70s¹⁷. We amended the manuscript by citing a review paper published in 2011¹⁸, reporting the overall hyalinization of the seminiferous tubule in infertile patients.

However, results on the basal lamina are novel and, to the best of our knowledge, our study is the first to clearly show that i) collagen I is over-deposited ii) while collagen IV is decreased, iii) seminiferous tubules with spermatogenesis and positive for collagen IV in ECM, colocalise in the

same parenchyma with seminiferous tubule without spermatogenesis and negative for collagen IV but over positive for collagen I, and iv) collagen organization undergone a visible remodelling (anisotropy, Figure 5F and 5G) in the hyalinized basal membrane of seminiferous tubules associated with germ cell aplasia.

We amended the manuscript as asked by the Reviewer by i) reporting deepened and validated characterization of the immature stage of Leydig cells, ii) removing sections about early ageing, and iii) citing the paper by Volkmann et al., 2011, with the overall hyalinization of the seminiferous tubule in infertile patients¹⁸.

Another concern is the over interpretation of results, which is outlined as one example for the Leydig cells below:

- The authors analysed testicular somatic cells from 3 patient with iGCA, lacking the germ cell population (SCO-phenotype) and compared results to normal data obtained from a sample with full spermatogenesis and published datasets. By analysis of SCO samples the authors indeed generated interesting datasets of testicular somatic cells, which are rarely represented in published datasets of testicular tissues with full spermatogenesis, in which the germ cells account for about 75% of the total cell population. However, a balanced number of normal and iGCA samples would have been desirable.

We thank the Reviewer for pointing to the novelty of our data on somatic cells.

In this revised version of the manuscript, we have added a second dataset of control testes to the Guo et al.² one (Sohni³ datasets); in new Figure 1H, we show that our pools consists of 4304 somatic cells from iGCA testes, a number that is rather similar to the 7637 somatic cells from control testes. Furthermore, the independent comparison of iGCA dataset with the one obtained in our lab from a CFTR-mutated patient (OA) gave results similar to those obtained in the comparison with Guo+Sohni datasets, now included as Supplementary Figure S16.

- For Leydig cells, authors conclude that they display an immature phenotype based on the 'overexpression' of the DLK1 gene. To prove the immaturity of Leydig cells compared to the normal situation, authors should have provided analyses also at protein level, comparing the two patient groups. Apart from that, they could have employed published data (Guo et al. 2019) for comparison with immature human Leydig cells obtained from prepubertal boys.

We agree with this Reviewer that the immaturity of the Leydig cells can play a central role in the NOA. In the original version of the manuscript, the immature state of Leydig cells in iNOA men was supported by transcriptional changes of the pair DLK1/NOTCH2, IGF2, INSL3, HSD17B3 genes and associated to a decreased blood level of Testosterone in iNOA vs age-matched fertile individuals.

As suggested by Reviewer 2, in the revised version of our study we further characterised Leydig cells immaturity at transcriptional level by comparing our datasets with those reported by Guo and Sohní that included data from neonatal and prepuberal testis^{2,4}. We now report that iGCA Leydig cells are stuck at an immature stage, with a signature recapitulating the stage A-B of maturation identified in neonatal-prepuberal human testis (see Figure 4 and Figure S20). In addition, the immature state of iGCA Leydig cells was supported by the reduced level of INSL3 protein in iGCA Leydig cells vs control testis. In the Supplementary Information we also add the IF detection of DLK1 and IGF2 in iGCA vs control adult testes showing the over-expression of DLK1 and the reduced expression of IGF2 at the protein level.

- As only transcriptional data is provided, the interpretation that 'Leydig cell immaturity, at least since puberty, may favour the boost of a chronic pro-inflammatory environment, which, on the one hand is responsible for the fibrosis of the seminiferous tubules and on the other eventually for germ cell depletion' is one example of an over interpretation of results. As the authors do not have any information whether germ cells were ever present in the testicular tissues of this patient group, the suggestion that germ cells were depleted is pure speculation. It would be important to compare the datasets with the available data from immature human testicular tissues to elucidate how the situation in these testicular tissues compares to normal development. In general, this study would greatly benefit from a more focused and in depth analyses of selected issues that are touched by this manuscript (facilitating essential validation and comparison to available datasets) and importantly an interpretation that is based on the datasets.

We agree with the Reviewer about the over-interpretation of our findings. The association between the occurrence of a chronic pro-inflammatory environment and germ cell depletion has been removed. The paragraph at page 18, lines 375-382 was removed from the Discussion.

This Reviewer is right in suggesting looking for the presence of the germ cells in immature vs mature testes. In fact, a recent paper confirmed the presence of germ cells also in the immature testes⁴.

In this revised version of our study, to demonstrate the immature status of Leydig cells in the iGCA testis, we have used several datasets:

- GSE124263, by Sohni et al. 2019, reporting data from adult and neonatal testes;
- GSE112013, by Guo et al. 2018, reporting data from 3 adult testes;
- GSE120506, by Guo et al. 2018, reporting data from 1 neonatal testis;
- GSE134144, by Guo et al. 2020, reporting data from 4 pre-puberal testes.

The list of these datasets has been included as Supplementary Table S36.

Reviewer #3 (Remarks to the Author):

In this manuscript the authors perform a detailed single cell RNA-Seq analysis of testis samples from men with idiopathic germ cell aplasia (iGCA) and one sample from a case of obstructive azoospermia (OA) that retains germ cells. Multiple different somatic cell types including immune cells were identified and transcriptional profiles compared to a previously published reference dataset from normal human testis. Differentially expressed genes in some of the somatic cell types of iGCA samples including inflammatory markers and genes associated with extracellular matrix. Based on gene expression analysis between the different cell types, many cell pathways were downregulated in most somatic cell types of iGCA samples including those associated with protein synthesis. Transcriptional profiles also suggested that hormone producing Leydig cells were developmental immature and this correlated with reduced testosterone levels in the circulation of infertile men with non-obstructive azoospermia (NOA). Analysis of testis sections from cohorts of NOA men confirmed changes in extracellular matrix deposition and suggested a chronic inflammatory response. Systemic levels of pro-inflammatory cytokines were also elevated in NOA patients. Combined, the authors propose that iNOA testis may develop as a result of chronic testicular (and systemic) inflammation and a premature ageing process within the testis somatic cells.

The manuscript will be of interest to the field and single cell datasets generated from iNOA samples invaluable. Mechanisms of most NOA cases are unclear. In general, the analysis has been performed in a careful and thorough manner and the identification of systemic indicators of iNOA very interesting. However, some points require further exploration and there are major concerns over the control samples used for the analysis.

We thank this reviewer for appreciating the potential impact of our study.

Major points:

1. The authors subject samples from iNOA and OA individuals to single cell RNA-Seq but no controls (fertile, age-matched men) were sampled and analysed in their manuscript. To assess changes in gene expression patterns in testis somatic cells from patients, they compare to samples previously analysed by the group of Bradley Cairns. However, more consideration needs to be given as to whether these are suitable as controls. Are they age matched? Was cell extraction and processing equivalent?

We agree with the Reviewer that we did not produced data for fertile age-matched men. Due to ethical restrictions, we cannot harvest testicular parenchyma from fertile men undergoing surgery for varicocele, testis torsion or other testicular diseases. The best testis parenchyma that we can use as control is the non-neoplastic tissue collected from orchiectomy. The main drawback of non-neoplastic tissue is that the neoplastic environment might alter the transcriptional profiles in the surrounding parenchyma, a problem that we cannot sort out. Therefore, we decided not to use such tissues.

For this reason, we used published data from control tissues obtained in non-European countries.

The included the Supplementary Table S36 lists all datasets used, including our own with data from 3 iGCA and 1 OA testis. Details show i) age of the donors, ii) sample processing and sequencing, and iii) how the data were used in our study.

We checked that in our and others studies similar procedures had been used for tissue dissociation, consisting of enzymatic digestion completed in 30 minutes. The cell processing was also similar, and the RNA-seq was performed with the 10x Genomic system on the Chromium platform

Collection and storage of the tissues were the main differences between our study and others we used. In our study, we used fresh testicular tissue obtained during microTESE intervention, and the tissues were kept in cold Krebs solution and processed within 15 minutes from the collection in the surgery room (as reported in the M&M section). In the studies from others, the authors used tissues from autoptic donors (Guo et al.) or collected after vasectomy (Sohni et al.). The tissue specimens were frozen and then thawed for RNAseq.

We agree that fresh vs autoptic or frozen tissues might give biased results in the differential gene expression and modulated pathways analyses. To overcome this technical limitation, we have firstly combined control donors from multiple input datasets (from Cairns' and Wilkinson's groups). Secondly, we have used the integration method available in Seurat v3, which is aimed at correcting for technical differences between datasets (i.e. batch effect correction), and performing comparative scRNA-seq analysis across different experimental conditions. Lastly, we added a comparative analysis of iNOA samples with the RNAseq dataset obtained from testis of an OA man with normal spermatogenesis processed in our lab. We found similar DEG and deregulated pathways in LEY and MYD cells as in the comparison with others' datasets. This analysis is now reported in the Supplementary Information, as Supplementary Figure S16.

In conclusion, we are confident that the comparisons made in this study are not biased by the tissue collection and storage methodologies.

Finally, since we are aware that age, but also ethnicity and environment, may affect gene transcription, for the validation analysis (presented in Figure 6), we used iNOA and control that were aged-matched, ethnicity-matched (white Caucasian) and environment-matched (north of Italy).

Concerns as to the suitability of the control data used is illustrated by the fact that many common cell pathways (48) are predicted to be downregulated in all different somatic cell types present in the iNOA testis, including many ribosomal protein genes. This seems very unlikely given the diversity of cell types within the testis and suggests that the controls are not suitable. The iNOA samples should also be compared to control datasets from other labs e.g. Miles Wilkinson. In addition, although the OA sample may not be ideal as a control, at least it was processed alongside the iNOA samples so a comparison between OA and iNOA samples would be very informative. Are similar pathways misregulated in iNOA somatic cells versus those from the OA patient?

We thank the Reviewer for suggesting this interesting way of controlling the setup of our work. We have now included the two control samples from Miles Wilkinson lab (Sohni et al.³) together with the Guo et al. samples² originally used as controls. All analyses have been re-run and results are presented in the new Figure panels (1G, 1H, all panels in Figure 2, all panels in Figure 3, all panels in Figure 4, 5A, 6L). We confirmed the downregulation of pathways related to protein metabolism (including selenocysteine synthesis and selenoamino acid metabolism) in iNOA samples compared to

Miles Wilkinson lab (Sohni et al.³), thus, confirming that the downregulation was not due to technical problems with the control samples in Guo et al.²

However, we cannot rule out the possibility that all control datasets available, including those suggested by the referee from Miles Wilkinson lab, are not suitable, due to the difficulties in obtaining such tissues. Therefore, following the referee's suggestion, we now provide an integration analysis of the 3 iNOA samples vs the OA sample as an additional control, which is now reported in the Supplementary Information, as Supplementary Figure S16. The UMAP of Figure S16, shows that most somatic cells from the OA patient mapped to iNOA LEY and MYD. When we compared the transcriptomes, LEY and MYD cells from the iNOA patients vs. the OA patient show upregulation of ECM organization, Scavenging by Class A receptor, Pre-NOTCH transcription and translation, and downregulation of ribosome biogenesis and Selenocysteine synthesis. These results overlap what we obtained in the comparison with Guo and Sohn CTLs^{2,3}, thus validating the quality of the other two datasets^{2,3}.

We thank this Reviewer again as the analysis of iNOA vs OA supports our interpretation of the results in the analysis of iNOA vs CTL.

2. T cells and fibroblasts are identified from iNOA testis but not present in the control single cell datasets used. It is unclear if this is because T cells and fibroblasts are only present in iNOA testis or they are just not represented in the Cairns dataset. Is this true for other available human testis single cell datasets? This limits the conclusions that can be drawn about these populations. Are there alternative single cell datasets of T cells and fibroblasts from other human tissues that can be used as comparison? For example, are there unique features of T cells present in iNOA testis versus other tissue-resident T cells?

We thank the Reviewer for raising this point. Several single-cell RNA seq databases were used to characterize the phenotype of the somatic cell populations resident in the iGCA testis, including T cells¹⁴ and stromal cells, previously identified as fibroblasts that now we identified as pericytes^{7,9}. The characterization of STRO and T cells has also been already commented on in reply to Reviewer 1. Please read the reply to Reviewer#1 (Specific comment 1).

3. Based on single cell analysis, Leydig cells are proposed to be functionally immature in iNOA samples, which correlated with reduced testosterone production. INSL3 is a maker of Leydig cell maturity and was analysed in iNOA Leydig cells. Confusingly, more cells were found to be INSL3 positive in iNOA samples than those from controls although of the cells that were positive in controls, the median expression level of INSL3 was higher than that of iNOA samples. Does this indicate that the control samples contained many more immature Leydig cells than the iNOA samples (unexpected) but that those cells which were mature were "more mature" than those from iNOA samples? This point needs further exploration and doesn't seem consistent with the broad conclusion that iNOA Leydig cells are more immature than controls. How do the iNOA Leydig cells compare to cells from pre-pubertal immature human testis?

We thank the Reviewer for raising this point, which was also highlighted by other Reviewers.

Please read Replies to Reviewer 1 (Major comments point 3 and Specific comment 7) and to Reviewer 2 (last paragraph) about immaturity of LEY cells.

4. Changes in expression of imprinted genes are found in iNOA somatic populations although the significance not explored. Are epigenetic regulators known to be involved in maintaining gene imprinting mis-expressed in iNOA cells? What is the potential functional significance of this observation?

We agree with this reviewer that the epigenetic regulation of imprinted genes is a potential factor to unveil the pathogenesis of iNOA, as well as to understand the time of the onset (i.e., starting during embryogenesis). At present, main epigenetic studies have been performed on spermatocyte genomes overlooking the somatic compartment.

Here we report the novel finding of a genomewide alteration of imprinting, in that several genes located on different chromosomes are differentially expressed, either up or down, in LEY and MYD cell populations, at least (i.e. Meg8, Kcnq1ot1).

Even more unexpected, a selection of either maternally or paternally imprinted genes is present in LEY and MYD transcriptomes from iGCAs while completely absent in control cells. The reverse also holds true with transcripts present in the CTLs populations and absent in iGCA (H19/Igf2, Dlk1, Hymai, to name a few).

A very upstream mechanism with genomewide effect might be involved. However, a very deep investigation would be necessary to reach a meaningful conclusion. We can anticipate that some good candidates from transcription profiles have been identified.

5. Histological analysis of iNOA testis indicated dynamic changes in collagen I (increased) and collagen IV (decreased). However, the single cell analysis seemed to show only increased expression of collagen subunits in somatic cells. How are these two observations reconciled?

In our study, DGE analyses showed upregulation of collagen I genes in LEY, MYD, SRT, END and STRO cells of iGCA, and upregulation of some Collagen IV genes in END, MYD and STRO cells. In contrast, IHC analyses did not show any collagen IV deposition in the seminiferous tubule, as expected by the gene expression data while, IHC showed a decrease in Collagen I deposition.

When collagen birefringence was measured, we noticed a disorganized topography of collagens in the iGCA seminiferous tubules suggesting that a post-translational control might change the balance of the deposited collagen isoforms.

We previously reported that the testis parenchyma of iGCAs showed an altered composition of extracellular matrix in the basal lamina of tubules (decreased amount of Laminin4, Laminin5, NID2 (Nidogen), HSPG2, and upregulation of OGN¹⁹. The formation of supramolecular Col IV aggregates and network assembly in the basal membrane is dependent on the laminin network via nidogen²⁰.

A potential interpretation of these information is that the impaired composition and tridimensional organization of ECM in the seminiferous tubules might be due to unbalanced concentrations of a protein ensemble in iGCA somatic tissues. Col IV isoforms, although overexpressed, might not be detectable because not assembled in the basal membrane.

Overview of interactions of basement membrane major molecules, from Mak et al.²⁰

6. Changes in the localisation pattern of HMGB1 are suggested to indicate cellular senescence and chronic inflammation in somatic cells of iNOA testis. Are other known indicators of cellular senescence (e.g. cell cycle-related proteins) and inflammatory signalling changed in iNOA samples? On a related note, the gH2A.X staining suggested DNA damage in iNOA cells but no signal was apparent in normal testis. Typically, meiotic germ cells should be gH2AX positive due to the DNA breaks specifically induced during that process. Are germ cells positive in the control samples? Are known components/targets of the DNA damage response differentially expressed in iNOA cells?

The Reviewer correctly suggested that meiotic germ cells should be positive for gH2AX due to Spo11 nuclease activity before chromosome crossing-over in pachytene and represent a positive control for staining.

As our IHC images had limited nuclear resolution, we applied IF staining with very good outcomes and appreciated the presence of gH2AX in meiotic spermatocytes. In the iGCA testes, accumulation of gH2AX was detected in the Leydig and Myoid cell, but not in the Sertoli cells as reported in our previous version of the manuscript. We are sorry for having previously shown a misleading information due to the technical approach.

We amended the section of Results, and placed gH2AX IF images in Figure 6B. The Discussion section has also been amended accordingly.

As asked by this Reviewer, we also verify the senescent phenotype of the somatic cells in the iGCA testis. Upon p16INK4a/CDKN2A staining, mainly Leydig cells showed a strong signal and can be considered senescent. This information is reported in the Supplementary Figure S22.

7. Changes in systemic levels of vitamin D are proposed to indicate enhanced ageing in iNOA individuals. Why have the authors focused specifically on vitamin D? Is this routinely used as a marker of the aging process in humans? Additional details on the utility of this marker should be provided.

Why have the authors focused specifically on vitamin D?

The rationale is that Vitamin D plays a direct role in ageing processes, controls systemic inflammation, oxidative stress and mitochondrial respiratory function and senescence²¹. Likewise, a recent study on 9,940 individuals reported that vitamin D declines during aging with a linear loss equal to 2.9 nmol/L each 10 years of age from the age of 50. In addition, vitamin D level is inversely associated with mortality^{22,23}.

In the comprehensive framework of diagnostic work-up for male infertility, vitamin D level is tested in candidates to assisted reproductive techniques. As it is evident from our Figure 6, many of our iNOA men are very low in VitD and need supplementation, although the protocol has not been implemented in Clinic.

Is this routinely used as a marker of the aging process in humans? *At present vitamin D is not routinely used as a marker of aging in humans; however, in the context of male infertility vitamin D assessment may add information on the redox status of the patients and can contribute to a better therapeutic management (i.e., supplement of vitamin D²⁴ or anti-oxidants²⁵).*

Additional details on the utility of this marker should be provided. *Although of potential interest, both in the framework of ageing and male infertility, it is still unknown if Vitamin D is an active or passive player. Therefore, after a thorough internal discussion we agreed to remove the data about Vitamin D and the relative discussion.*

8. A major issue concerns the over-arching conclusion of the study – that chronic inflammation and other changes in testis somatic cells (senescence etc) are responsible for loss of germ cells in iNOA patients. However, as samples are not analysed at earlier stages during disease progression, can it be excluded that the observed changes in somatic cells are a result rather than a cause of the loss of germ cells at various stages of testis development/maturation? In essence, without functional data supporting a causative role for the described changes in somatic cells in germ cell aplasia, this conclusion cannot be made. The discussion needs to be modified accordingly.

The over-arching conclusions of the study, such as “responsibility of chronic inflammation for germ cell loss”, or that “early ageing of iNOA men depends on deregulated transcriptional pathways” have been removed.

Indeed, we elaborated on tissue ageing because of i) a SASP phenotype, ii) an excess of p16/CDKN2A in LEY cells, iii) an active DNA Damage Response and iv) local high levels of inflammatory cells and mediators (HMGB1).

Therefore, at the end of the Discussion session, we conclude the manuscript as follows:

“This study unveils novel insights into idiopathic germ cell aplasia, such the immature phenotype of LEY cells stuck at the prepuberal stage of development and expressing clear markers of senescence that might support the hypothesis of exhaustion in the somatic progenitor cell compartment, still needing an experimental validation. Pathways associated to mechanisms potentially responsible for the iGCA testis phenotypes open new perspectives for a better understanding of the pathobiology behind the clinical condition of idiopathic non-obstructive azoospermia and the potential association with the clinical manifestations as chronic non-oncological diseases and higher rates of cancers.”

Minor points

1. Lines 340-341 – to exclude the possibility that endothelial cells play a role in germ cell aplasia seems premature given that they are proposed play important roles in rodent germ cell maintenance and they do have changes in gene expression versus control. Functional data would be required to exclude this possibility.

Thanks for pointing out our overstatement. Because we did not validate the statement, we removed the sentence “Therefore, we may exclude any potential pathogenic contribution of END cells in the context of iGCA”.

2. Lines 375-377 – effects of inflammatory cytokines on stem cell function are discussed. However, the quoted references concern hematopoietic and intestinal stem cells. Any evidence that they have a similar effect on germline stem cells?

We thank the Reviewer for pointing out the mistake. Indeed, in agreement with his/her comment #8 about over-arching conclusion, from the Discussion section we have removed the entire speculation not proven by data. The section on inflammatory cytokines effect on germinal stem cell that was removed is at page 18, lines 375-382.

REFERENCES

- 1 Wang, M. *et al.* Single-Cell RNA Sequencing Analysis Reveals Sequential Cell Fate Transition during Human Spermatogenesis. *Cell Stem Cell* **23**, 599-614 e594, doi:10.1016/j.stem.2018.08.007 (2018).
- 2 Guo, J. *et al.* The adult human testis transcriptional cell atlas. *Cell Res* **28**, 1141-1157, doi:10.1038/s41422-018-0099-2 (2018).
- 3 Sohni, A. *et al.* The Neonatal and Adult Human Testis Defined at the Single-Cell Level. *Cell Rep* **26**, 1501-1517 e1504, doi:10.1016/j.celrep.2019.01.045 (2019).
- 4 Guo, J. *et al.* The Dynamic Transcriptional Cell Atlas of Testis Development during Human Puberty. *Cell Stem Cell* **26**, 262-276 e264, doi:10.1016/j.stem.2019.12.005 (2020).
- 5 Heng, T. S., Painter, M. W. & Immunological Genome Project, C. The Immunological Genome Project: networks of gene expression in immune cells. *Nat Immunol* **9**, 1091-1094, doi:10.1038/ni1008-1091 (2008).
- 6 Malhotra, D. *et al.* Transcriptional profiling of stroma from inflamed and resting lymph nodes defines immunological hallmarks. *Nat Immunol* **13**, 499-510, doi:10.1038/ni.2262 (2012).
- 7 Muhl, L. *et al.* Single-cell analysis uncovers fibroblast heterogeneity and criteria for fibroblast and mural cell identification and discrimination. *Nat Commun* **11**, 3953, doi:10.1038/s41467-020-17740-1 (2020).
- 8 Paquet-Fifield, S. *et al.* A role for pericytes as microenvironmental regulators of human skin tissue regeneration. *J Clin Invest* **119**, 2795-2806, doi:10.1172/JCI38535 (2009).
- 9 Tabib, T., Morse, C., Wang, T., Chen, W. & Lafyatis, R. SFRP2/DPP4 and FMO1/LSP1 Define Major Fibroblast Populations in Human Skin. *J Invest Dermatol* **138**, 802-810, doi:10.1016/j.jid.2017.09.045 (2018).
- 10 Davidoff, M. S. The Pluripotent Microvascular Pericytes Are the Adult Stem Cells Even in the Testis. *Adv Exp Med Biol* **1122**, 235-267, doi:10.1007/978-3-030-11093-2_13 (2019).
- 11 Kumar, D. L. & DeFalco, T. A perivascular niche for multipotent progenitors in the fetal testis. *Nat Commun* **9**, 4519, doi:10.1038/s41467-018-06996-3 (2018).
- 12 Jiang, M. H. *et al.* Characterization of Nestin-positive stem Leydig cells as a potential source for the treatment of testicular Leydig cell dysfunction. *Cell Res* **24**, 1466-1485, doi:10.1038/cr.2014.149 (2014).
- 13 Shami, A. N. *et al.* Single-Cell RNA Sequencing of Human, Macaque, and Mouse Testes Uncovers Conserved and Divergent Features of Mammalian Spermatogenesis. *Dev Cell* **54**, 529-547 e512, doi:10.1016/j.devcel.2020.05.010 (2020).
- 14 Szabo, P. A. *et al.* Single-cell transcriptomics of human T cells reveals tissue and activation signatures in health and disease. *Nat Commun* **10**, 4706, doi:10.1038/s41467-019-12464-3 (2019).
- 15 Efremova, M., Vento-Tormo, M., Teichmann, S. A. & Vento-Tormo, R. CellPhoneDB: inferring cell-cell communication from combined expression of multi-subunit ligand-receptor complexes. *Nat Protoc* **15**, 1484-1506, doi:10.1038/s41596-020-0292-x (2020).
- 16 Wu, F. C. *et al.* Hypothalamic-pituitary-testicular axis disruptions in older men are differentially linked to age and modifiable risk factors: the European Male Aging Study. *J Clin Endocrinol Metab* **93**, 2737-2745, doi:10.1210/jc.2007-1972 (2008).
- 17 de Kretser, D. M., Kerr, J. B. & Paulsen, C. A. The peritubular tissue in the normal and pathological human testis. An ultrastructural study. *Biol Reprod* **12**, 317-324, doi:10.1095/biolreprod12.3.317 (1975).
- 18 Volkmann, J. *et al.* Disturbed spermatogenesis associated with thickened lamina propria of seminiferous tubules is not caused by dedifferentiation of myofibroblasts. *Hum Reprod* **26**, 1450-1461, doi:10.1093/humrep/der077 (2011).
- 19 Alfano, M. *et al.* Impaired testicular signaling of vitamin A and vitamin K contributes to the aberrant composition of the extracellular matrix in idiopathic germ cell aplasia. *Fertil Steril* **111**, 687-698, doi:10.1016/j.fertnstert.2018.12.002 (2019).

- 20 Mak, K. M. & Mei, R. Basement Membrane Type IV Collagen and Laminin: An Overview of
Their Biology and Value as Fibrosis Biomarkers of Liver Disease. *Anat Rec (Hoboken)* **300**,
1371-1390, doi:10.1002/ar.23567 (2017).
- 21 Wimalawansa, S. J. Vitamin D Deficiency: Effects on Oxidative Stress, Epigenetics, Gene
Regulation, and Aging. *Biology (Basel)* **8**, doi:10.3390/biology8020030 (2019).
- 22 Schottker, B. *et al.* Serum 25-Hydroxyvitamin D Levels as an Aging Marker: Strong
Associations With Age and All-Cause Mortality Independent From Telomere Length,
Epigenetic Age Acceleration, and 8-Isoprostane Levels. *J Gerontol A Biol Sci Med Sci* **74**, 121-
128, doi:10.1093/gerona/gly253 (2019).
- 23 Schottker, B. *et al.* Vitamin D and mortality: meta-analysis of individual participant data from
a large consortium of cohort studies from Europe and the United States. *BMJ* **348**, g3656,
doi:10.1136/bmj.g3656 (2014).
- 24 Cito, G. *et al.* Vitamin D and Male Fertility: An Updated Review. *World J Mens Health* **38**,
164-177, doi:10.5534/wjmh.190057 (2020).
- 25 Martin-Hidalgo, D., Bragado, M. J., Batista, A. R., Oliveira, P. F. & Alves, M. G. Antioxidants
and Male Fertility: from Molecular Studies to Clinical Evidence. *Antioxidants (Basel)* **8**,
doi:10.3390/antiox8040089 (2019).

Reviewers' Comments:

Reviewer #1:

Remarks to the Author:

Comments:

The author has reversed his/her manuscript carefully. More powerful evidence has been provided and most of my comments have been explained or reversed. However, there are still some questions and concerns that I hope to be addressed.

1. P33 line 681-684 : the author has added clinical information in the reversed manuscript. I found the patient #2 seem to show a normal level of LH (7 mUI/ml), whether it was the reason that patient #2 showed a larger heterogenic transcriptome (e.g. no END was captured) with the two other iNOA? Histological analysis of testis parenchyma for all the three iNOA patient are necessary.

2. The author has repeated gH2AX analyses using the immunofluorescence protocol and found accumulation of gH2AX in Leydig and Myoid cells of iGCA, but not in the Sertoli cells. However, the image was not clear and no statistics were provided.

3. Line 273: "The higher tumor frequency reported for iGCA subjects" needs a reference to support.

Line 285 "Myoid and Leydig nuclei were negative in control (CTL) testis but stained positive in iGCA tissues". This figure was not clear to show the expression of gH2AX. Could the author change the pseudo-color of chromatin to BLUE? Here is a gH2AX(Anti-phospho-Histone H2A.X(Ser139) Antibody, Millipore, Cat#05-636; RRID:AB_309864) IHC staining of OA testis by our lab.

4. Line 285 "In addition, by DAPI staining, we noted very distorted nuclei in the LEY and SRT cell population". How did the author make sure that these distorted nuclei belong to LEY. SRT is easy to identify because it is the only cell type left in seminiferous tubules of iNOA patients, but in the "interstitial space" of the testis, there are also other cell types such as mast cells and macrophages and these cells showed high-level proliferation in iNOA patients. So some bio-markers of LEY should be staining here to further confirm this result.

5. In the reversed manuscript, the author showed some evidence that the immaturity of the Leydig cells plays a central role in the NOA. However, he/she also found a strong accumulation of p16, a global senescence-associated gene, in iGCA LEY cells. Could he/she give some explanation for this paradox?

6. In Zhao's and Guo's papers, they annotated the STRO cluster as smooth muscle or vascular smooth muscle, and here is an image of FABP4 (a STRO marker from table s11) IHC from HumanAtlas database.

7. The immaturity of Leydig cells is an interesting observation. However, unlike prepuberty, iNOA patients often showed a high level of LH. So, it may be caused by a downstream pathway. The author should discuss the clinical value and significance of this interesting observation, such as 1) is the immaturity of Leydig cells a cause or effect of the loss of germ cells (and the immaturity of Sertoli cells)? 2) if there are some potential treatments to improve the maturity arrest of LEY cells? 3) if IGF2 is a potential predictor for TESE?

Reviewer #2:

Remarks to the Author:

Massimo Alfano and colleagues have submitted a revised version of their manuscript 'Human idiopathic germ cell aplasia is characterized by aged and inflamed testis somatic components with an active DNA damage response. The authors have addressed the comments made and thereby further improved the manuscript. They have largely removed overstatements from this manuscript. One point of concern that remains is the following comment in the introduction 'NOA can be one of the first manifestations of early ageing (line 55). As the loss of germ cells is by no means an effect of healthy aging (Pohl et al.2019, Laurentino et al., 2020) - this part of the sentence is not supported by available data and needs to be revised.

Apart from that, the authors need to revise the figures, so that the figure legend and text

throughout the figure becomes readable. Examples are Figure 1E,F; Figures 2A-D (blue to red scale is not readable, nor are the legends for J. Also for Figure 3A, there would be space to increase the letter size, so that the pathways can be properly seen. Finally, Figure 6 (right panel) is difficult to decipher. Apart from these aspects I have no further comments.

Reviewer #3:

Remarks to the Author:

The authors have made considerable effort to address concerns. In particular, by analysis of additional control testis datasets and improved immunostaining analysis. The text has also been revised quite extensively. As a result the manuscript is much improved and will be of great interest to the field. All major concerns have been addressed.

Remaining minor points:

1. It would be useful to readers to highlight the differences in processing of testis samples for the authors dataset and the published control samples as discussed in the rebuttal (frozen vs fresh etc). These methodological details can be important for interpretation of data and future studies.
2. As a result of the comprehensive analysis performed by the authors, the manuscript now contains 35 supplementary tables and 25 supplementary figures. This extended supplementary dataset is a bit overwhelming and the figures sometimes only contain one or two data panels. Can some of the suppl figures and/or tables be consolidated? Are they all necessary?

Reply letter (manuscript NCOMMS-20-35263A)

Reviewer #1 (Remarks to the Author):

The author has reversed his/her manuscript carefully. More powerful evidence has been provided and most of my comments have been explained or reversed. However, there are still some questions and concerns that I hope to be addressed. We thank the Reviewer#1 for the overall positive comment to the revised version of our manuscript.

1. P33 line 681-684: the author has added clinical information in the reversed manuscript. I found the patient #2 seem to show a normal level of LH (7 mUI/ml), whether it was the reason that patient #2 showed a larger heterogenic transcriptome (e.g. no END was captured) with the two other iNOA? Histological analysis of testis parenchyma for all the three iNOA patient are necessary.

We agree with the Reviewer #1 that a LH value of 7 mUI/ml would seem normal, but patient #2 correctly fitted into the primary (non-compensated¹) hypogonadism group from the clinical standpoint, with a semen analysis phenotype suggestive for idiopathic non obstructive azoospermia. Histological analysis of the testis parenchyma from the three iNOA men with germ cell aplasia have been provided; a similar pattern was found in all cases. The three hematoxylin-eosin stains have been included in the Supplementary Figure S1 and the related legend amended accordingly.

2. The author has repeated gH2AX analyses using the immunofluorescence protocol and found accumulation of gH2AX in Leydig and Myoid cells of iGCA, but not in the Sertoli cells. However, the image was not clear and no statistics were provided.

According to the suggestion of the Reviewer#1, we have verified the specificity of the commercial anti-gamma H2AX monoclonal antibody Millipore, Cat#05-636 on HeLa cells treated with H2O2 (Supplementary Figure S12). Therefore, we have repeated the analysis of control and iGCA testes, thus providing high quality images. As discussed throughout the main text, we observed gamma H2AX staining only on meiosis in normal tissues, while up to 35% of SRT, MYD and LEY cells stained positive for gammaH2AX in iGCA testes. Because of the absence of positive signal in the somatic cells of controls, it was not possible to perform a statistical analysis.

The manuscript has been amended with high quality images and the text has been revised accordingly.

3. Line 273: “The higher tumor frequency reported for iGCA subjects” needs a reference to support.

We thank the Reviewer#1 for this relevant criticism. Indeed it was a mistake, since we wanted to refer to azoospermia, which in many cases is associated with GCA. The text has been rephrased, as follows “---The higher tumor frequency reported for men with azoospermia, prompted us to proceed from chronic tissue inflammation to senescence and DNA damage as possible causes of genomic instability in iGCA...”. Accordingly,

we did not quote a new reference as the statement had been already referenced in the Introduction section (refs 11 and 12).

4. Line 285 “Myoid and Leydig nuclei were negative in control (CTL) testis but stained positive in iGCA tissues”. This figure was not clear to show the expression of gH2AX. Could the author change the pseudo-color of chromatin to BLUE? Here is a gH2AX (Anti-phospho-Histone H2A.X(Ser139) Antibody, Millipore, Cat#05-636; RRID:AB_309864) IHC staining of OA testis by our lab. As reported in Comment #2, we have repeated the analysis using the blue pseudo-color to show chromatin (both for Figure 6B and the gH2AX control staining in Supplementary Figure S12).

We confirm the use of the monoclonal gH2AX Ab; likewise, the staining on meiosis is in agreement with the image provided by the Reviewer#1. Therof, we thank the Reviewer#1 since the overall quality of our imaging was improved and fully supported novel foreground in the condition of iGCA (see reply to comment #2).

5. Line 285 “In addition, by DAPI staining, we noted very distorted nuclei in the LEY and SRT cell population”. How did the author make sure that these distorted nuclei belong to LEY. SRT is easy to identify because it is the only cell type left in seminiferous tubules of iNOA patients, but in the “interstitial space” of the testis, there are also other cell types such as mast cells and macrophages and these cells showed high-level proliferation in iNOA patients. So some bio-markers of LEY should be staining here to further confirm this result.

In the legend of Figure 6C, we have mentioned that parameters of nuclear distortion have been quantified on cells immunostained for a lineage specific marker of LEY cells; in this context, we referred to Figure S7 and S11 that depict the Leydig specific marker CALB2.

6. In the reversed manuscript, the author showed some evidence that the

immaturity of the Leydig cells plays a central role in the NOA. However, he/she also found a strong accumulation of p16, a global senescence-associated gene, in iGCA LEY cells. Could he/she give some explanation for this paradox?

Cellular senescence is a multifaceted process induced in response to a variety of stresses and physiological signals which may cause genomic or epigenomic damage — or metabolic deficits — all of which potentially putting cells at risk for oncogenic transformation. Two hallmarks of cellular senescence are an essentially irreversible arrest of cell proliferation and the development of a multi-component senescence-associated secretory phenotype (SASP). Unlike apoptotic and quiescent cells, senescent cells are highly metabolically active, and this process promotes several activities, including local inflammation^{2,3}.

We certainly agree with the Reviewer#1 that immaturity of Leydig cells associated with senescent phenotype seems a paradox or, and even better, a sort of loop argument like the “chicken and the egg”.

In this context, our interpretation poses that LEY cells conserve an immature phenotype up to the adult life and are exposed to a constant pressure of stimulatory inputs. The inability of LEY cells to respond to inputs and mature impedes the resolution of the signaling chain that turns to stress and senescence. As a suggested hypothesis, a chronic unbalanced high FSH levels along with low testosterone levels might represent stress signals eventually resulting in an unbalanced release of molecules altering feedback mechanisms in the pituitary/Sertoli/Leydig loop. We did not discuss this further interpretation throughout the manuscript to avoid overstatements in absence of supporting data (as asked by all Reviewers in the previous submission).

This interesting paradox is indeed a hot topic⁴⁻⁶ and would require further studies in the next future, but as clearly outlined by the Reviewer#1, the information reported in our study opens up avenues for new investigations.

7. In Zhao’s and Guo’s papers, they annotated the STRO cluster as smooth muscle or vascular smooth muscle, and here is an image of FABP4 (a STRO marker from table s11) IHC from HumanAtlas database.

We amended the text, by referring the STRO cluster as pericytes/ vascular smooth muscle cells and adding the related references to Zhao’s and Guo’s paper.

8. The immaturity of Leydig cells is an interesting observation. However, unlike prepuberty, iNOA patients often showed a high level of LH. So, it may be caused by a downstream pathway. The author should discuss the clinical value and significance of this interesting observation, such as 1) is the immaturity of Leydig

cells a cause or effect of the loss of germ cells (and the immaturity of Sertoli cells)? 2) if there are some potential treatments to improve the maturity arrest of LEY cells? 3) if IGF2 is a potential predictor for TESE?

Please find individual answers to the latter questions.

1. It is known that the somatic environment regulates the viability of the germ cells; it has been reported that knock-out of the retinoic acid degrading enzyme *Cyp26b1* in the Sertoli cells leads to an accumulation of retinoic acid in the embryonic testis, which is associated with the absence of germ cells and a lower testis volume in neonates⁷. If the immaturity of the Leydig cell is the cause or, vice versa, the effect of germ loss is a hypothesis which certainly deserves further investigations.

2. In preclinical models, several pollutants (i.e., Bisphenol B, Dibutyltin, Phthalates, Trimethyltin) have shown to inhibit maturation of Leydig cells, as well as testosterone and sperm production⁸⁻¹¹. On the contrary, 2,2',4,4'-Tetrabromodiphenyl ether or rhGH-treatment may favor the maturation of immature Leydig cells, the production of testosterone and the spermatogenesis^{12,13}. Resveratrol supplementation has also been shown to increase spermatogenesis and testosterone production in adult animals¹⁴, and sperm number and motility in patients with idiopathic infertility¹⁵, while reducing sperm DNA fragmentation of TID patients¹⁶. However, all the above studies imply that germ cells are present in the seminiferous tubules and that Leydig cells are not senescent.

Here we report the accumulation of collagen I and a decreased amount of collagen IV in the basal lamina of the seminiferous tubule of iGCA. We hypothesize that the hyalinization of the basal lamina due to the accumulation of collagen I is the result of the chronic inflammatory environment. We also recently reported an impaired biochemical composition of the the basal lamina of the seminiferous tubule of iGCA¹⁷. The functionally impaired composition of the hyalinized basal lamina might represent one of the causes of germ cells loss, which could not engage with the basal lamina through adhesion molecules/integrins, thus undergoing apoptosis.

According to what discussed at point 6, that i) immature Leydig cells undergo senescence upon chronic stimuli, and ii) the senescent status is responsible for the local inflammation, we might speculate that a control of the local inflammation – either the intensity or the chronicity – might reduce the damage of the basal lamina, thus favoring the survival of germ cells. The potential impact of the hyalinization of the basal lamina of the seminiferous tubule and the presence/absence of germ cell has been reported in Figure 5 and discussed in the Discussion section of the manuscript.

Potentially, senolytic treatments³ might be of use to arrest/revert the impaired testis environment associated with iGCA by reducing senescence in LEY cells and the downstream pathways such as inflammation and fibrosis. Indeed, senolytic treatments have recently entered in several clinical trials on patient with chronic inflammatory diseases¹⁸, including those with idiopathic pulmonary fibrosis¹⁹. Conversely, there is not yet an effective application in patients with iNOA.

The discussion concerning the potential role of the chronic environment has been removed in the last version of the manuscript following Reviewers' comments. Indeed, we agree that the above speculations are overstatements, although the novel outcomes

provided by our study are shedding light on new potential mechanisms and treatments that warrant confirmation in future projects.

3. IGF2 is expressed by Leydig cells but also by organs such as the liver. To the best of our knowledge, we are not aware if IGF2 can be used as predictive marker of sperm retrieval in NOA patients undergoing TESE. We have measured the peripheral level of IGF2 in non-Finnish white-European iNOA vs. age-matched, same race, fertile men and did not find significant information.

- 1 Tajar, A. *et al.* Characteristics of secondary, primary, and compensated hypogonadism in aging men: evidence from the European Male Ageing Study. *J Clin Endocrinol Metab* **95**, 1810-1818, doi:10.1210/jc.2009-1796 (2010).
- 2 Wiley, C. D. & Campisi, J. From Ancient Pathways to Aging Cells-Connecting Metabolism and Cellular Senescence. *Cell Metab* **23**, 1013-1021, doi:10.1016/j.cmet.2016.05.010 (2016).
- 3 Wiley, C. D. *et al.* Oxylin biosynthesis reinforces cellular senescence and allows detection of senolysis. *Cell Metab* **33**, 1124-1136 e1125, doi:10.1016/j.cmet.2021.03.008 (2021).
- 4 Menendez, J. A. & Alarcon, T. Senescence-Inflammatory Regulation of Reparative Cellular Reprogramming in Aging and Cancer. *Front Cell Dev Biol* **5**, 49, doi:10.3389/fcell.2017.00049 (2017).
- 5 Borghesan, M., Hoogaars, W. M. H., Varela-Eirin, M., Talma, N. & Demaria, M. A Senescence-Centric View of Aging: Implications for Longevity and Disease. *Trends Cell Biol* **30**, 777-791, doi:10.1016/j.tcb.2020.07.002 (2020).
- 6 Di Micco, R., Krizhanovsky, V., Baker, D. & d'Adda di Fagagna, F. Cellular senescence in ageing: from mechanisms to therapeutic opportunities. *Nat Rev Mol Cell Biol* **22**, 75-95, doi:10.1038/s41580-020-00314-w (2021).
- 7 MacLean, G., Li, H., Metzger, D., Chambon, P. & Petkovich, M. Apoptotic extinction of germ cells in testes of Cyp26b1 knockout mice. *Endocrinology* **148**, 4560-4567, doi:10.1210/en.2007-0492 (2007).
- 8 Li, Y. *et al.* Bisphenol B stimulates Leydig cell proliferation but inhibits maturation in late pubertal rats. *Food Chem Toxicol* **153**, 112248,

- doi:10.1016/j.fct.2021.112248 (2021).
- 9 Li, G., Chang, X., Zhao, Y., Li, D. & Kang, X. Dibutyltin (DBT) inhibits in vitro androgen biosynthesis of rat immature Leydig cells. *Toxicology* **456**, 152779, doi:10.1016/j.tox.2021.152779 (2021).
- 10 Zhu, X. *et al.* Exposure to di-n-octyl phthalate during puberty induces hypergonadotropic hypogonadism caused by Leydig cell hyperplasia but reduced steroidogenic function in male rats. *Ecotoxicol Environ Saf* **208**, 111432, doi:10.1016/j.ecoenv.2020.111432 (2021).
- 11 Hu, J., Zhang, D., Yan, Z. & Cheng, Y. The in vitro effects of trimethyltin on the androgen biosynthesis of rat immature Leydig cells. *Toxicology* **444**, 152577, doi:10.1016/j.tox.2020.152577 (2020).
- 12 Li, Z. *et al.* Low dose of fire retardant, 2,2',4,4'-tetrabromodiphenyl ether (BDE47), stimulates the proliferation and differentiation of progenitor Leydig cells of male rats during prepuberty. *Toxicol Lett* **342**, 6-19, doi:10.1016/j.toxlet.2021.02.006 (2021).
- 13 Huh, K., Nah, W. H., Xu, Y., Park, M. J. & Gye, M. C. Effects of Recombinant Human Growth Hormone on the Onset of Puberty, Leydig Cell Differentiation, Spermatogenesis and Hypothalamic KISS1 Expression in Immature Male Rats. *World J Mens Health* **39**, 381-388, doi:10.5534/wjmh.200152 (2021).
- 14 Juan, M. E. *et al.* trans-Resveratrol, a natural antioxidant from grapes, increases sperm output in healthy rats. *J Nutr* **135**, 757-760, doi:10.1093/jn/135.4.757 (2005).
- 15 Illiano, E. *et al.* Resveratrol-Based Multivitamin Supplement Increases Sperm Concentration and Motility in Idiopathic Male Infertility: A Pilot Clinical Study. *J Clin Med* **9**, doi:10.3390/jcm9124017 (2020).
- 16 Simas, J. N., Mendes, T. B., Fischer, L. W., Vendramini, V. & Miraglia, S. M. Resveratrol improves sperm DNA quality and reproductive capacity in type 1 diabetes. *Andrology* **9**, 384-399, doi:10.1111/andr.12891 (2021).
- 17 Alfano, M. *et al.* Impaired testicular signaling of vitamin A and vitamin K contributes to the aberrant composition of the extracellular matrix in idiopathic germ cell aplasia. *Fertil Steril* **111**, 687-698, doi:10.1016/j.fertnstert.2018.12.002 (2019).
- 18 Kirkland, J. L. & Tchkonja, T. Senolytic drugs: from discovery to translation. *J Intern Med* **288**, 518-536, doi:10.1111/joim.13141 (2020).
- 19 Justice, J. N. *et al.* Senolytics in idiopathic pulmonary fibrosis: Results from a first-in-human, open-label, pilot study. *EBioMedicine* **40**, 554-563, doi:10.1016/j.ebiom.2018.12.052 (2019).

Reviewer #2 (Remarks to the Author):

Massimo Alfano and colleagues have submitted a revised version of their manuscript 'Human idiopathic germ cell aplasia is characterized by aged and inflamed testis somatic components with an active DNA damage response. The authors have addressed the comments made and thereby further improved the manuscript. They have largely removed overstatements from this manuscript. One point of concern that remains is the following comment in the introduction 'NOA can be one of the first manifestations of early ageing (line 55). As the loss of germ cells is by no means an effect of healthy aging (Pohl et al.2019, Laurentino et al., 2020) - this part of the sentence is not supported by available data and needs to be revised.

The text has been revised accordingly.

Apart from that, the authors need to revise the figures, so that the figure legend and text throughout the figure becomes readable. Examples are Figure 1E,F; Figures 2A-D (blue to red scale is not readable, nor are the legends for J. Also for Figure 3A, there would be space to increase the letter size, so that the pathways can be properly seen. Finally, Figure 6 (right panel) is difficult to decipher. Apart from these aspects I have no further comments.

Figures 1, 2, 3 and 6 have been amended.

Reviewer #3 (Remarks to the Author):

The authors have made considerable effort to address concerns. In particular, by analysis of additional control testis datasets and improved immunostaining analysis. The text has also been revised quite extensively. As a result the manuscript is much improved and will be of great interest to the field. All major concerns have been addressed.

We thank the Reviewer#3 for the overall positive comment to the revised version of our manuscript.

Remaining minor points:

1. It would be useful to readers to highlight the differences in processing of testis samples for the authors dataset and the published control samples as discussed in the rebuttal (frozen vs fresh etc). These methodological details can be important for interpretation of data and future studies.

We thank the Reviewer#3 for this insightful comment. Thereof, we have provided methodological details in a new paragraph of the Methods section “Testis datasets used in this study: similarities and differences”.

2. As a result of the comprehensive analysis performed by the authors, the manuscript now contains 35 supplementary tables and 25 supplementary figures. This extended supplementary dataset is a bit overwhelming and the figures sometimes only contain one or two data panels. Can some of the suppl figures and/or tables be consolidated? Are they all necessary?

Once more, we thank the Reviewer#3 for this suggestion. Accordingly, we have consolidated several Supplementary Figures, which turned to 13.

Reviewers' Comments:

Reviewer #1:

Remarks to the Author:

No further comments. Congratulation !

REVIEWERS' COMMENTS

Reviewer #1 (Remarks to the Author):

No further comments. Congratulation !

All the authors of the manuscript want to take the opportunity provided by this letter to thank the Reviewers for their constructive comments that have made possible to improve the quality and relevance of this study.